# A combined cryo-EM and molecular dynamics approach reveals the mechanism of ErmBL-mediated translation arrest

Stefan Arenz[1,*], Lars V. Bock[2,*], Michael Graf[1], C. Axel Innis[3,4,5], Roland Beckmann[1,6], Helmut Grubmüller[2], Andrea C. Vaiana[2] & Daniel N. Wilson[1,6]

Nascent polypeptides can induce ribosome stalling, regulating downstream genes. Stalling of ErmBL peptide translation in the presence of the macrolide antibiotic erythromycin leads to resistance in *Streptococcus sanguis*. To reveal this stalling mechanism we obtained 3.6-Å-resolution cryo-EM structures of ErmBL-stalled ribosomes with erythromycin. The nascent peptide adopts an unusual conformation with the C-terminal Asp10 side chain in a previously unseen rotated position. Together with molecular dynamics simulations, the structures indicate that peptide-bond formation is inhibited by displacement of the peptidyl-tRNA A76 ribose from its canonical position, and by non-productive interactions of the A-tRNA Lys11 side chain with the A-site crevice. These two effects combine to perturb peptide-bond formation by increasing the distance between the attacking Lys11 amine and the Asp10 carbonyl carbon. The interplay between drug, peptide and ribosome uncovered here also provides insight into the fundamental mechanism of peptide-bond formation.

[1] Gene Center and Department for Biochemistry, University of Munich, Feodor-Lynenstrasse 25, Munich 81377, Germany. [2] Department of Theoretical and Computational Biophysics, Max Planck Institute for Biophysical Chemistry, Am Fassberg 11, Göttingen 37079, Germany. [3] Institut Européen de Chimie et Biologie, University of Bordeaux, Pessac 33607, France. [4] INSERM U1212, Bordeaux 33076, France. [5] CNRS UMR7377, Bordeaux 33076, France. [6] Center for integrated Protein Science Munich, University of Munich, Feodor-Lynenstrasse 25, Munich 81377, Germany. * These authors contributed equally to this work. Correspondence and requests for materials should be addressed to A.C.V. (email: andrea.vaiana@mpibpc.mpg.de) or to D.N.W. (email: wilson@lmb.uni-muenchen.de).

Ribosomes are the protein-synthesizing machines of the cell. The active site for peptide-bond formation, the peptidyl-transferase centre (PTC), is located on the large ribosomal subunit[1]. During translation elongation, peptide-bond formation occurs when a peptidyl-tRNA is located at the P-site and an aminoacyl-tRNA at the A-site of the PTC. The α-amino group of the aminoacyl-tRNA makes a nucleophilic attack on the carbonyl carbon of the peptidyl-tRNA, transferring the peptidyl moiety on the A-tRNA and prolonging the nascent polypeptide chain by one amino acid[1]. As the nascent polypeptide chain is synthesized, it passes through a tunnel located on the large ribosomal subunit and emerges on the solvent side where protein folding occurs[2]. A growing body of evidence has revealed that the ribosomal tunnel is not a passive conduit for the nascent polypeptide chain but rather plays an important role in protein folding, co-translational targeting and translation regulation[2].

Translation can be regulated by intrinsic properties of the nascent chain, such that the amino-acid sequence of the polypeptide is sufficient to modulate its rate of synthesis, and in some cases even induce translation arrest[3]. Well-characterized bacterial examples include the MifM and SecM leader peptides, which interact with the ribosomal tunnel to induce translational stalling and thereby regulate expression of a downstream gene in the operon[3]. Nascent polypeptide-mediated translation regulation can also require the presence of an additional ligand or cofactor[3–5]. Well-characterized examples include the TnaC leader peptide, which requires the presence of free tryptophan for stalling to occur[3], and the Erm leader peptides, which induce translation stalling in the presence of the macrolide antibiotics[4,5].

In the case of Erm peptides, translational arrest on the leader peptide leads to expression of the downstream macrolide resistance determinant, usually a methyltransferase that confers resistance by methylating A2058, a 23S ribosomal RNA (rRNA) nucleotide that comprises part of the macrolide-binding site within the ribosomal tunnel. For example, in *Streptococcus sanguis* the expression of the ErmB methyltransferase is controlled by programmed arrest during translation of the upstream *ermBL* leader peptide[6,7] (Fig. 1a). In the absence of erythromycin (ERY), ErmB expression is repressed because the ribosome-binding site (RBS) and AUG start codon of the *ermB* mRNA are sequestered in a stem–loop structure (Fig. 1a). However, in the presence of sub-inhibitory concentrations of ERY, ribosomes translating the ErmBL leader peptide become stalled, leading to an alternative stem–loop structure in the mRNA that exposes the RBS and start codon of the *ermB* gene and thus allows ribosome binding and induction of ErmB expression (Fig. 1a). The drug-dependent nature of the stalling ensures that expression only occurs when the drug is present, which is critical for survival because of the fitness cost associated with the methylation of A2058 (ref. 8).

Previous studies demonstrated that polymerization of the ErmBL nascent chain halts because the ribosome is unable to catalyse peptide-bond formation between the 10 amino-acid-long ErmBL-tRNA$^{Asp}$ (codon 10) in the ribosomal P-site and Lys-tRNA$^{Lys}$ (codon 11) in the A-site[9] (Fig. 1a). The ErmBL-dependent translational arrest occurs with both macrolides (for example, ERY) and ketolides (for example, telithromycin)[9], which contrasts with other leader peptides, such as ErmCL, where stalling is specific for macrolides[5,10]. Mutagenesis studies identified four residues within ErmBL (Arg7 and Val9-Lys11) that are critical for stalling[9] and two of which (Arg7 and Asp10) play a role in determining the antibiotic specificity of ErmBL[11]. Interestingly, the compromised stalling efficiency of the ErmBL-R7A mutant can be rescued by compensatory mutations of 23S nucleotide U2586, located within the ribosomal tunnel, but not by mutations in the neighbouring tunnel nucleotide A2062

that impair ErmCL stalling[10,12]. The first structural insights into how the ErmBL leader peptide, the macrolide antibiotic and the ribosomal tunnel cooperate to inactivate the PTC of the ribosome were derived from a cryo-electron microscopy (cryo-EM) structure of the ErmBL-stalled ribosomal complex (ErmBL-SRC) at 4.5–6.6-Å resolution[9]. On the basis of this structure, it was suggested that peptide-bond formation cannot occur because the ErmBL nascent chain stabilizes a non-productive state of the ribosome that prevents accommodation of Lys-tRNA at the A-site.

Here we present cryo-EM structures of the ErmBL-SRC in the presence and absence of A-tRNA at 3.6-Å resolution, which were used as a basis to perform extensive molecular dynamics (MD) simulations. Our analyses revealed a complex network of hydrogen-bonding interactions between the C-terminal region of ErmBL and nucleotides of the 23S rRNA, consistent with the importance of these residues for translational stalling. Unexpectedly, the C-terminal Asp10 residue of ErmBL was observed in a rotated conformation, promoting a unique path of the nascent chain such that it bypasses the tunnel-bound macrolide. In the cryo-EM structure, the unusual conformation of ErmBL appears to distort the terminal A76 ribose of the P-tRNA, which is supported by MD simulations performed after removal of ERY, showing that the ribose distortion is alleviated as the ErmBL nascent chain moves into the volume previously occupied by the drug. In addition, MD simulations predict that the presence of ERY also influences the position of the aminoacyl moiety of the A-tRNA via a cascade of 23S rRNA rearrangements. Thus, our results illustrate how ErmBL and ERY cooperate to perturb both the A- and P-tRNA positions at the PTC to inhibit peptide-bond formation and induce translational arrest.

## Results

**Cryo-EM structures of the ErmBL-SRC.** The ErmBL-SRCs were generated by translation of a dicistronic *2XermBL* mRNA in the presence of 20 μM ERY using an *Escherichia coli* lysate-based *in vitro* translation system. The ErmBL-SRC disomes were then isolated by sucrose gradient purification, converted to monosomes and analysed using single-particle cryo-EM (see Methods). *In silico* sorting of cryo-EM images yielded two major homogeneous subpopulations of ribosomes bearing P-tRNA and E-tRNAs, but differing in the presence (termed ErmBL-APE-SRC; 85,393 particles, 33%) or absence (termed ErmBL-PE-SRC; 75,839 particles, 30%) of A-tRNA (Fig. 1b and Supplementary Fig. 1). Both subpopulations could be refined to an average resolution of 3.6 Å (using a Fourier shell correlation (FSC) cutoff of 0.143; Supplementary Fig. 2), with local resolution calculations indicating that the ribosomal core reaches towards 3.1 Å (Fig. 1c and Supplementary Fig. 2). At this resolution, base separation is clearly observed for rRNA nucleotides (Fig. 1d), as is the density for the majority of the amino-acid side chains in ribosomal proteins (Fig. 1e). Density is also observed for the lysyl moiety attached to the CCA end of the A-tRNA as well as for the ErmBL leader peptide attached to CCA end of the P-tRNA (Fig. 1f).

**A molecular model for the ErmBL nascent chain.** The ErmBL nascent chains of both ErmBL-SRCs were well resolved within the ribosomal tunnel, as indicated by local resolution (Supplementary Fig. 3). Moreover, the electron density for ErmBL in the ErmBL-PE-SRC was essentially identical to that in the ErmBL-APE-SRC (Supplementary Fig. 3), indicating that the presence of the A-tRNA does not significantly influence the conformation of the ErmBL nascent chain. With the exception of the side chain of Asn8 for which we observe no density, we are able to present a

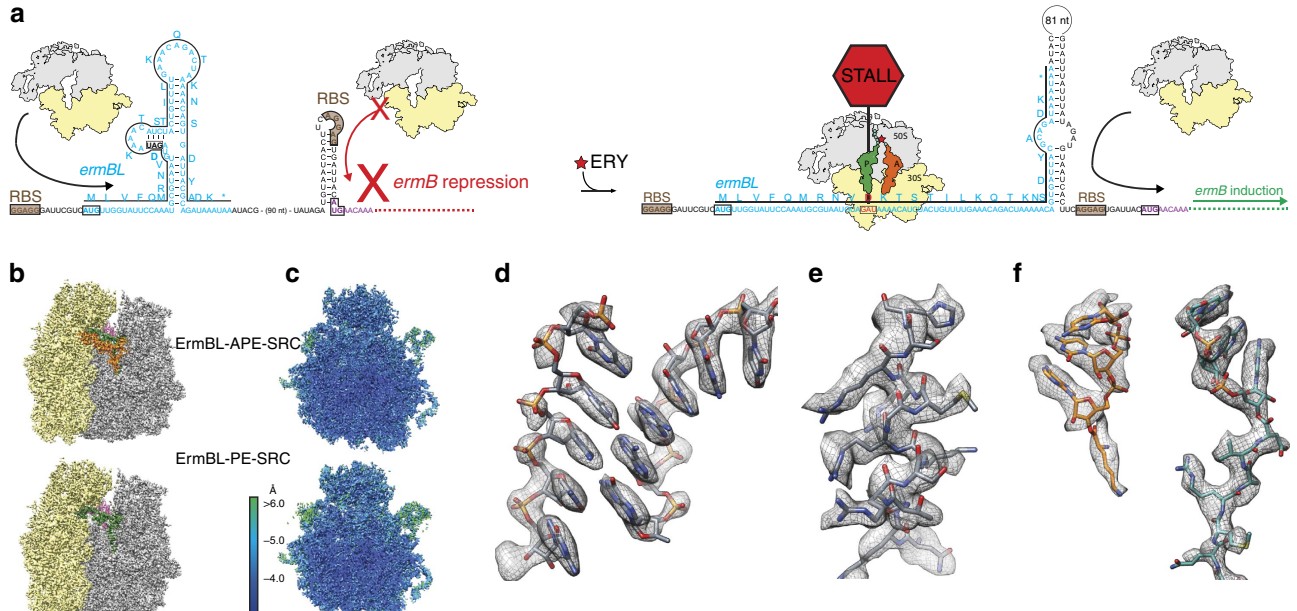

**Figure 1 | Cryo-EM structure of the ErmBL-SRC.** (**a**) Schematic for *ermBL*-dependent regulation of *ermB* translation in the presence of ERY. (**b**) Cryo-EM maps of the ErmBL-APE-SRC and ErmBL-PE-SRC, with 30S (yellow), 50S (grey), A-tRNA (orange), P-tRNA (green) and E-tRNA (magenta). (**c**) Transverse sections of the cryo-EM maps of the ErmBL-APE-SRC and ErmBL-PE-SRC, coloured according to local resolution. (**d**–**f**) Examples of electron density (grey mesh) of the ErmBL-APE-SRC map illustrating (**d**) base separation within an rRNA helix, (**e**) side chains of ribosomal protein α-helix and (**f**) aminoacylated-CCA end of the A-tRNA (orange) and ErmBL peptide attached to the CCA end of the P-tRNA (green).

complete model for the ErmBL nascent chain residues Val3–Asp10 (Supplementary Table 1). In contrast, the N-terminus of ErmBL appears to be flexible, consistent with the reduced local resolution of this region, thus preventing the N-terminal Met1 residue from being modelled and allowing only the backbone for Leu2 to be tentatively assigned (Fig. 2a and Supplementary Fig. 3). The overall backbone trace of the ErmBL nascent chain within the ribosomal tunnel reported here is similar to the path reported previously for ErmBL based on a 4.5–6.6-Å cryo-EM map[9] (Supplementary Fig. 4). Specifically, ErmBL adopts a unique conformation that enables it to bypass the macrolide ERY bound within the tunnel without even contacting it (Fig. 2a). This finding is consistent with biochemical studies demonstrating that ErmBL stalling occurs with a variety of macrolides and thus appears insensitive to the chemical nature of the drug[9,11]. The ErmBL path is distinct from that reported for the drug-dependent ErmCL-stalling peptide[13], which establishes intimate interactions with the drug (Fig. 2b), consistent with the findings that ErmCL stalling occurs with macrolide antibiotics containing a C3-cladinose sugar, such as ERY, but not with ketolides, such as telithromycin that lack the C3-cladinose[5,10]. This structure contrasts with the available structures of other polypeptide chains within the ribosomal tunnel[14–20], which revealed paths that would be sterically obstructed by the presence of the macrolide, as exemplified here by the TnaC nascent chain[14] (Fig. 2c). As noted[9], the path of ErmBL illustrates the principle as to how specific amino-acid sequences allow translation of a subset of proteins, even in the presence of a macrolide-obstructed tunnel[21,22].

**MD simulations of the ErmBL-SRC.** To capture the structural behaviour and energetics of the ErmBL peptide in the ribosome and to understand how these are influenced by ERY, all-atom explicit-solvent MD simulations were carried out. The simulations were started from two initial structures: (i) the ErmBL-APE-

SRC structure (referred to as +ERY) and (ii) the ErmBL-APE-SRC structure after computational removal of ERY (−ERY). From each of these structures, two independent simulations were started. All simulations of the fully solvated complexes were set up using a previously established protocol[23,24] and carried out using GROMACS[25], the amber99sb force field[26] and the SPC/E water model[27]. The root mean square deviation (r.m.s.d.) of the simulations from the cryo-EM structure remained low (r.m.s.d. < 5.2 Å; Supplementary Fig. 5, total simulation time ~12 μs) in comparison with other, shorter MD simulations of ribosomes started from high-resolution X-ray structures (r.m.s.d. 6–10 Å)[23,28]. This underscores the quality of our simulation set-up and force field as well as that of the starting structure. Removal of ERY leads to increased dynamics and flexibility of the ErmBL nascent chain, in particular the N-terminal residues 2–5 of the ErmBL peptide (Fig. 2d), as seen by the root mean square fluctuations (r.m.s.f.'s; Fig. 2e). The absence of the drug allows the ErmBL nascent chain to explore the lumen of the tunnel, and to move into the volume otherwise occupied by the drug (Fig. 2d–f), consistent with the idea that ERY promotes the unique conformation of the ErmBL nascent chain.

**Interactions of ErmBL within the ribosomal tunnel.** In contrast to the previous reconstruction of ErmBL[9], the significantly higher resolution of the ErmBL-SRCs reported here permits a detailed analysis of the interactions of ErmBL with components of the ribosomal tunnel (Fig. 3a). The majority of the interactions are localized within the last four C-terminal amino acids of ErmBL, namely Arg7–Asp10, which collectively establish a total of nine potential hydrogen-bond interactions with nucleotides of the 23S rRNA: Asp10 can form three hydrogen bonds, one from the backbone carboxyl oxygen with the N3 of U2585 and two from the Asp10 side chain oxygen atoms with the N1 and/or N2 of G2061 as well as the N3 of U2506 (Fig. 3b). These interactions are likely to be important since most mutations of Asp10 abolish

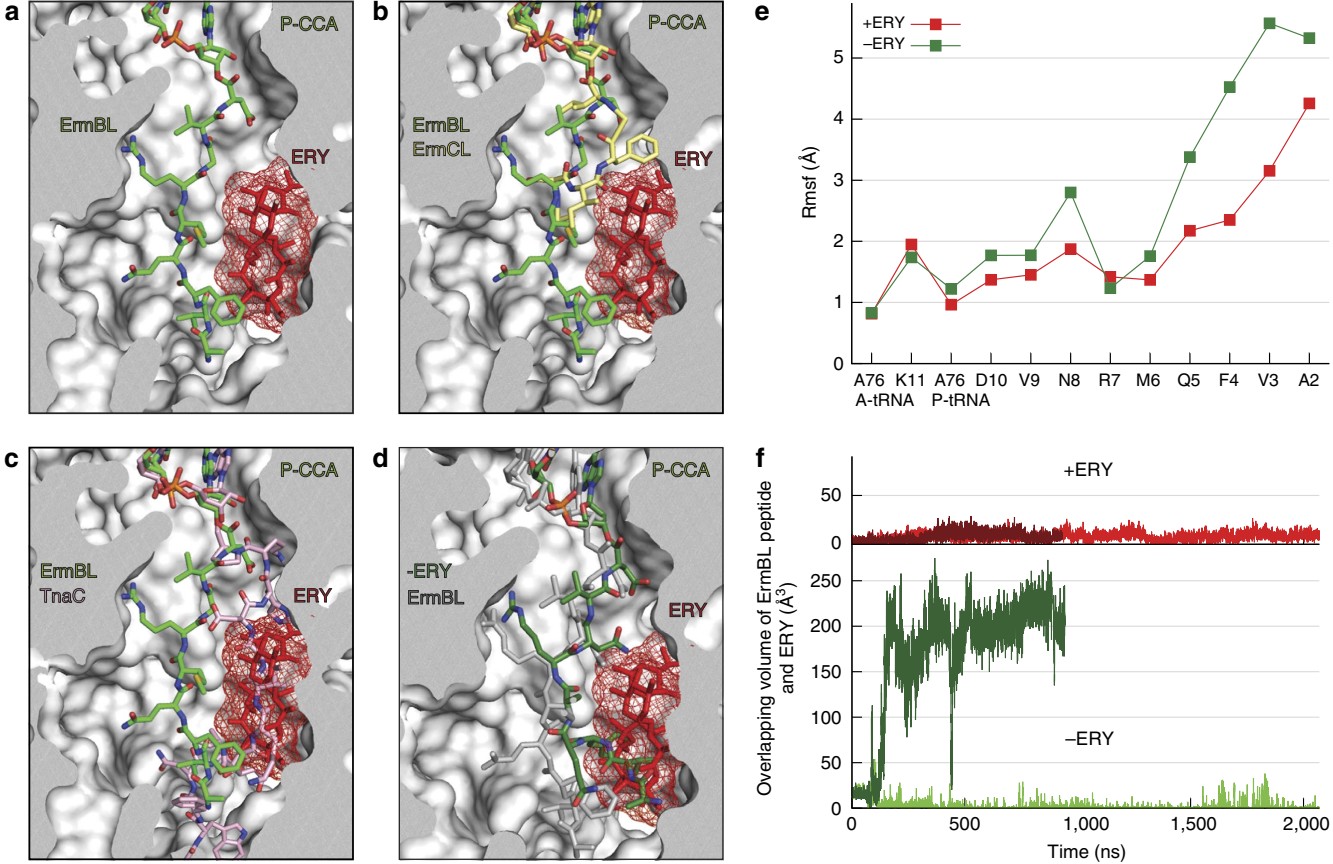

**Figure 2 | Path of ErmBL within the ribosomal tunnel.** (**a,b**) Path of (**a**) ErmBL (green) and (**b**) ErmCL (yellow) within the ribosomal tunnel with bound ERY (red mesh). (**c**) Path of TnaC (pink) compared with the path of ErmBL (green) within the ribosomal tunnel with superimposed ERY (red mesh). (**d**) Path of ErmBL after 700-ns simulation time in absence of ERY (green) compared with ErmBL as seen in the cryo-EM structure (grey). (**e**) Fluctuations of the tRNA A76 nucleotides and ErmBL residues in presence (red) and absence (green) of ERY obtained from the simulations. (**f**) Overlap between the ErmBL peptide and the volume occupied by ERY in the course of the simulations.

translational stalling[9,11]. All these interactions were also observed throughout the MD simulations. In particular, the hydrogen bonds between Asp10 and G2061 were very stable in all the simulations (62% average occupancy; Supplementary Fig. 6), with the formation of additional hydrogen bonds also being seen between the amide group of Asp10 and 23S rRNA nucleotide C2063 (74% average occupancy; Supplementary Fig. 6). The backbone carboxyl of Asn8 is within hydrogen-bonding distance of the ribose 2′OH of A2062, whereas no density is observed for the side chain, consistent with the report that Asn8Ala mutations do not affect ErmBL stalling[9]. In this respect, we note that the most favoured rotamers for the Asn side chain would sterically clash with the desosamine sugar of ERY. In addition, in the simulations with ERY, although the backbone dihedral angles of Asn8 remain almost constant (Supplementary Fig. 7), the side chain of Asn8 is more mobile than the neighbouring residues (Fig. 2e). Indeed, the Asn8 side chain adopts multiple conformations with transient hydrogen bonding to U2506, A2062 and C2063 (Supplementary Fig. 6), thus explaining the lack of density for this side chain (Fig. 3c).

The well-resolved density (Fig. 3d) and low mobility (Fig. 2e) for the Arg7 side chain can be explained by the multiple interactions that this positively charged residue establishes by protruding into a negatively charged rRNA pocket of the tunnel wall (Fig. 3d and Supplementary Fig. 6). Specifically, two hydrogen bonds are possible with the two terminal amino groups of Arg7, namely to the phosphate oxygen atoms of C2063 and

U2441 (Fig. 3d), which were also observed in the MD simulations (with an average occupancy of 13% and 87%, respectively, Supplementary Fig. 6). Additional hydrogen bonding is possible to the bridging ribose oxygen of A2439 and the ribose 2′OH of U2586 as well as between the backbone of NH of Arg7 and the O2 of U2586, although these interactions were not seen in the MD simulations (Supplementary Fig. 6). While an Arg7Ala mutation abolishes stalling[9,11], compensatory mutations of U2586 were shown previously to restore stalling[9], supporting the intimate association between these two entities. What is unclear from the structure is why the Val9Ala mutation also abolishes ErmBL-stalling activity[9]. In the ErmBL-SRC, the side chain of Val9 is well resolved, yet it does not appear to come into close proximity of any components of the ribosomal tunnel (Fig. 3e). We speculate that the Val9 side chain may, therefore, be important for indirectly promoting the correct conformation of the ErmBL nascent chain: one possibility is that Val9 provides some steric constraints that are necessary for the Arg7 side chain to be optimally oriented, since these two residues are close to each other within the ribosomal tunnel. Interestingly, in the absence of ERY, all the other peptide residues exhibit larger fluctuations than those seen in the simulations with ERY. The only exception is Arg7, the fluctuations of which are unchanged. This observation, together with the fact that the interaction enthalpies of Arg7 with the tunnel wall remain essentially unchanged upon removal of the antibiotic, suggests that the free energy barrier for detaching the Arg7 from the tunnel-bound conformation might be rate-limiting

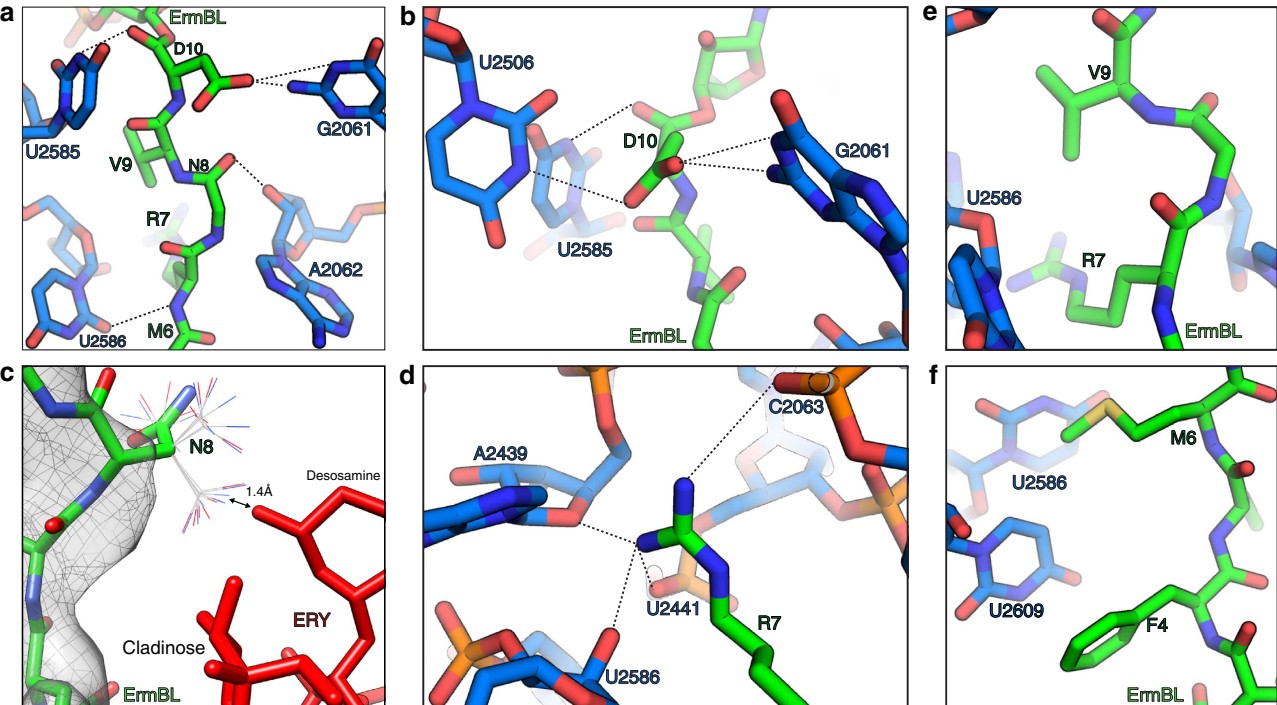

**Figure 3 | Interaction of the ErmBL nascent chain within the ribosomal tunnel.** (**a**–**f**) Interactions of ErmBL (green) with 23S rRNA nucleotides (blue). In **c**, different rotamers of the N8 side chain are illustrated and ERY is shown in red.

for full detachment of the peptide from the tunnel wall. Apparently, this barrier is too high to be overcome on the microsecond simulation timescale.

In contrast to the C terminus, the N-terminal half of ErmBL (residues Leu2–Met6) is not critical for stalling and each residue can be individually mutated to alanine without loss of activity[9]. This observation is consistent with both the decreased local resolution in this region (Supplementary Fig. 3) as well as with the large fluctuations of these residues observed in the MD simulations (Fig. 2e), indicating that the N terminus of ErmBL is flexible and does not adopt a single defined conformation. In addition, the sequence of the N-terminal MLVFQM is very hydrophobic and therefore provides few possibilities for hydrogen-bonding interactions. Instead, the Phe4 and Met6 side chains appear to pack against the U2609-A752 and U1782-U2586 base pairs, respectively (Fig. 3f).

**Rotation of Asp10 and an alternate path for ErmBL.** To understand how interactions of ErmBL within the ribosomal tunnel lead to translational arrest by preventing peptide-bond formation with the incoming Lys-tRNA in the A-site, we compared the ErmBL-SRC with available X-ray crystallography structures of ribosomal states with aminoacylated-tRNAs in the P-site. To date, these comprise structures of the *Haloarcula marismortui* 50S subunits bearing CC-puromycin (CC-pmn) or CC-Phe-caproic-acid-biotin tRNA analogues[29–31] and of *Thermus thermophilus* 70S ribosomes bearing fMet-tRNA[32] or Phe-tRNA[33]. In every single structure, the side chain of the Phe or fMet has the same orientation relative to the P-tRNA (Fig. 4a,b), namely pointing towards the direction of A2062. Unexpectedly, the Asp10 side chain attached to the P-tRNA in the ErmBL-SRC is rotated by 180° around the ester bond that links it to the 3′OH of A76, relative to the Phe/fMet side chains in the crystal structures (Fig. 4a,b). This unusual orientation might of course simply be due to the different respective tRNAs—a

peptidyl-tRNA for the ErmBL structure versus an aminoacyl-tRNA in the other crystal structures. To investigate this possibility, we also compared the ErmBL-SRC with structures of other ribosomal states bearing peptidyl-tRNAs in the P-site. There are currently four such cryo-EM structures available, the resolutions of which allow the orientation of the amino acid attached to the P-tRNA to be determined: TnaC-SRC[14], ErmCL-SRC[13], Sec61beta[20] and MifM-SRC[18]. In all four structures, the amino-acid side chain attached to the P-tRNA is oriented similar to the Phe/fMet side chain in the crystal structures (Fig. 4c–f). This finding suggests that the difference is not due to different peptides on the P-tRNA. In addition, differences in species are likely to be irrelevant here, since the ErmCL- and TnaC-SRC were also formed using *E. coli* ribosomes[13,14]. Finally, the rotation is also unlikely to be due to the nature of the amino acid because the residue that is attached to the P-tRNA in the MifM-SRC is also an Asp[18]. These findings lead us to conclude that the 180° rotation of the Asp10 residue in the ErmBL-SRC is likely to be a consequence of its context within the ErmBL nascent chain. It would therefore seem logical to assume that the interactions that stabilize the ErmBL in a specific conformation cooperatively stabilize the rotated state of Asp10.

**A-tRNA accommodation at the PTC of the ErmBL-SRC.** To understand in more detail how the unorthodox pathway of the ErmBL nascent chain could lead to inhibition of peptidyl-transferase activity, we compared the PTC of the ErmBL-SRC with crystal structures of the ribosome in different states of peptide-bond formation[29–32]. Initially, we focused on the A-tRNA, since it was previously proposed that ErmBL stabilized an unaccommodated Lys-tRNA at the A-site of the PTC[9]. In contrast to the previous study, the higher resolution achieved here enables us to more accurately place the CCA end of the A-tRNA and also to model the lysyl moiety (Fig. 5a), which was not visible before[9]. Comparison of the position of the CCA

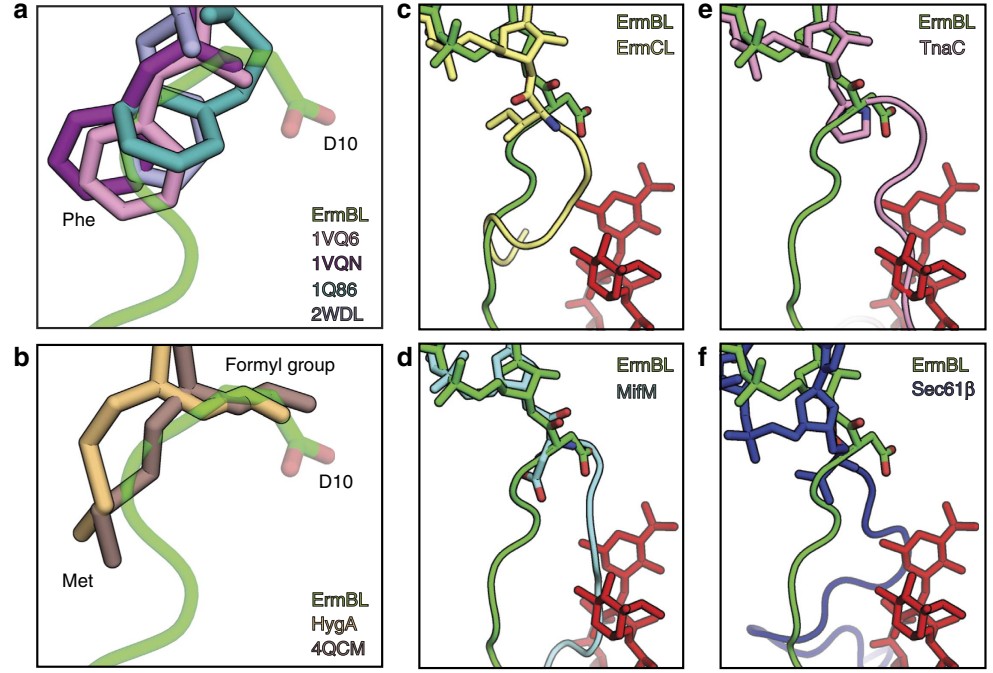

**Figure 4 | Rotation of the C-terminal amino acid and an alternate path of ErmBL.** (**a,b**) Orientation of ErmBL D10 compared with (**a**) CC-Phe-caproic-acid-biotin (CC-pcb) tRNA analogues (pink, PDB1VQ6; purple, PDB1VQN[30]; and teal, PDB1Q86 (ref. 74)), and Phe-tRNA (blue, PDB2WDL[33]) and with (**b**) *T. thermophilus* 70S ribosomes bearing fMet-tRNA (brown, PDB4QCM[32] and yellow, PDB 4Z3R[75]). (**c–f**) Alternate path of ErmBL (green) compared with (**c**) that of ErmCL (yellow, PDB 3J7Z[13]), (**d**) MifM (light blue, PDB3J9W[18]), (**e**) TnaC (pink, PDB4UY8 (ref. 14)) and (**f**) Sec61β (dark blue, PDB 3J92 (ref. 20)) in relation to the position of ERY (red).

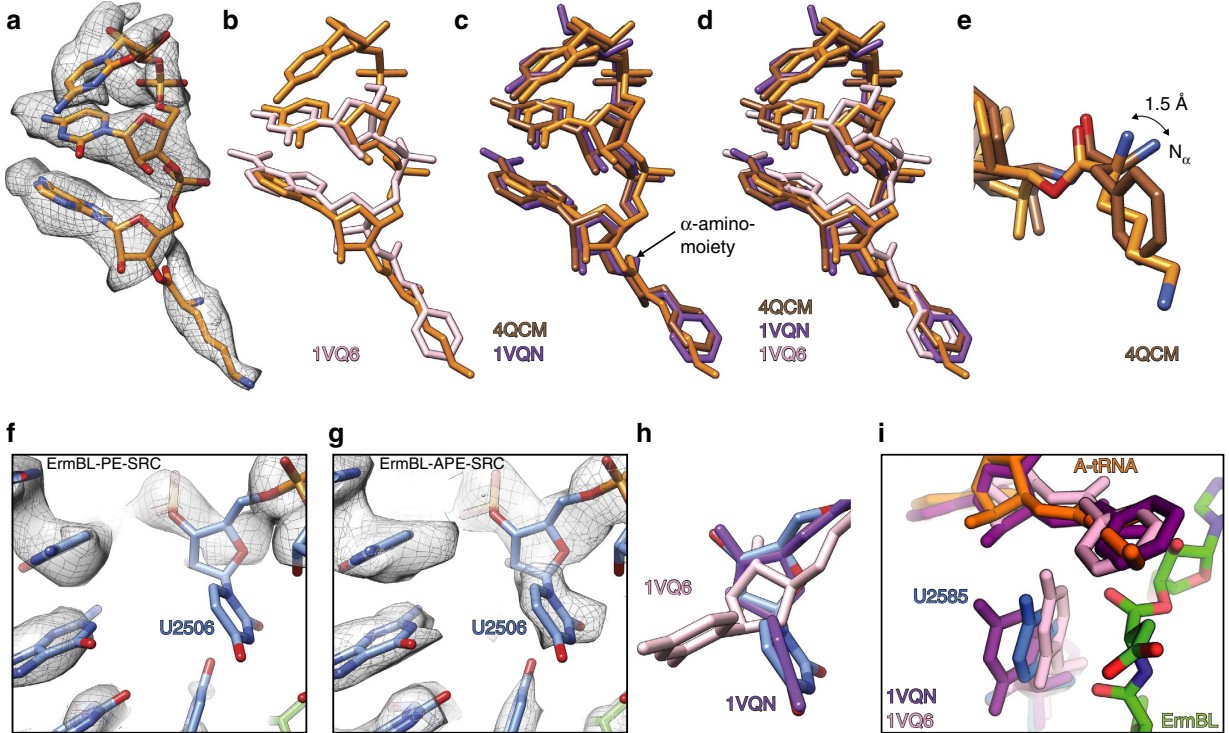

**Figure 5 | A-site tRNA accommodation at the PTC of the ErmBL-APE-SRC.** (**a–d**) Atomic model for the CCA end of the A-site tRNA (orange) and (**a**) electron density (grey mesh) in the ErmBL-APE-SRC compared with (**b**) unaccommodated (pink, PDB1VQ6 (ref. 30)) and (**c**) accommodated states (brown, PDB4QCM[32]; purple, PDB1VQN[30]) and (**d**) with both. (**e**) Positions of the α-amino group of the amino acid attached to the A-site tRNA in the ErmBL-APE-SRC compared with its position in the pre-peptide-bond formation state[32]. (**f–h**) Position of 23S rRNA nucleotide U2506 (blue) in (**f**) ErmBL-PE-SRC and (**g**) ErmBL-APE-SRC including electron density (grey mesh) and (**h**) compared with the unaccommodated and accommodated states of the PTC[30]. (**i**) Comparison of 23S rRNA nucleotide U2585 in ErmBL-APE-SRC (blue), compared with the unaccommodated and accommodated states of the PTC[30]. The nascent chain and A-tRNA in ErmBL are shown in green and orange, respectively.

end of the Lys-tRNA in ErmBL-SRC with the unaccommodated C-pmn and accommodated CC-pmn on the *H. marismortui* 50S subunit[29–31], as well as with accommodated CCA end of the Phe-tRNA in the *T. thermophilus* pre-attack complexes[32,33], suggests that the Lys-tRNA in the ErmBL-SRC is in a fully or near-fully accommodated state (Fig. 5b–e). The A-tRNA is shifted by 1.3 Å between the unaccommodated and accommodated states, when measuring the displacement using the ribose O4′ atom. In contrast, the Lys-tRNA in the ErmBL-SRC is shifted by only 0.9 and 0.6 Å relative to the unaccommodated and accommodated states, respectively.

Accommodation of the A-tRNA is accompanied by corresponding conformational changes within the 23S rRNA nucleotides of the PTC; specifically, shifts in U2506 and U2584-U2585 are thought to be diagnostic for A-tRNA binding[29–31]. In the ErmBL-PE-SRC, there is no clear density for U2506 (Fig. 5f). However, upon binding of Lys-tRNA to the A-site, as in the ErmBL-APE-SRC, U2506 adopts a well-defined conformation (Fig. 5g), which is similar to that observed in the accommodated state (Fig. 5h). In both the ErmBL-SRC structures, the density for U2585 is weaker than other PTC nucleotides, but appears to predominantly adopt an intermediate position between the accommodated and unaccommodated U2585 states (Fig. 5i), hinting that the Lys-tRNA is not fully accommodated as suggested previously[9]. We note however that regardless of whether the Lys-tRNA is fully or near-fully accommodated, in the refined model of the Lys-tRNA in the ErmBL-APE-SRC α-amino group is shifted by 1.5 Å compared with the accommodated pre-attack A-tRNAs (arrowed in Fig. 5e). The presence of the lysyl moiety on the A-tRNA indicates that the

nucleophilic attack has not occurred, consistent with the previous biochemical evidence indicating that the lysine amino acid does not get incorporated into the ErmBL nascent chain[9].

**Perturbation of the P-tRNA prevents peptide-bond formation.** Comparison of the CCA end of the P-tRNA in the ErmBL-SRC (Fig. 6a,b) with the accommodated position of CC-pmn[30], Phe-tRNA[33] or fMet-tRNA[32] reveals a significant shift by 1–2 Å of the ribose and nucleobase of A76 (Fig. 6c,d), which was also seen at lower resolution[9]. We note that the distance for the nucleophilic attack of the α-amino group of the Lys-tRNA on the carbonyl carbon of the P-tRNA is 4.1 Å in the ErmBL-SRC (Fig. 6e), whereas the equivalent distances in the pre-attack complexes are 3.2–3.3 Å (refs 32,33; Fig. 6f), which may contribute to the inhibition of peptide-bond formation observed in the ErmBL-SRC. Consistently, the conformation of the P-tRNA A76 ribose was most similar to that of the pre-attack state[32] in the MD simulations where ERY was absent (green traces in Fig. 6g), as compared with when the antibiotic was present (red traces in Fig. 6g). This finding suggests that the conformation of ErmBL induced by the presence of ERY in the ribosomal tunnel is responsible for the perturbed position of the P-tRNA.

The most recent model for peptide-bond formation, based on high-resolution X-ray crystallography structures, suggests that for the nucleophilic attack to proceed, a proton needs to be transferred from the attacking nucleophile to a water molecule (W1) via a proton wire formed by the 2′ OH of the P-site A76 ribose and the 2′OH of A2451 (ref. 32; Fig. 6h and Supplementary

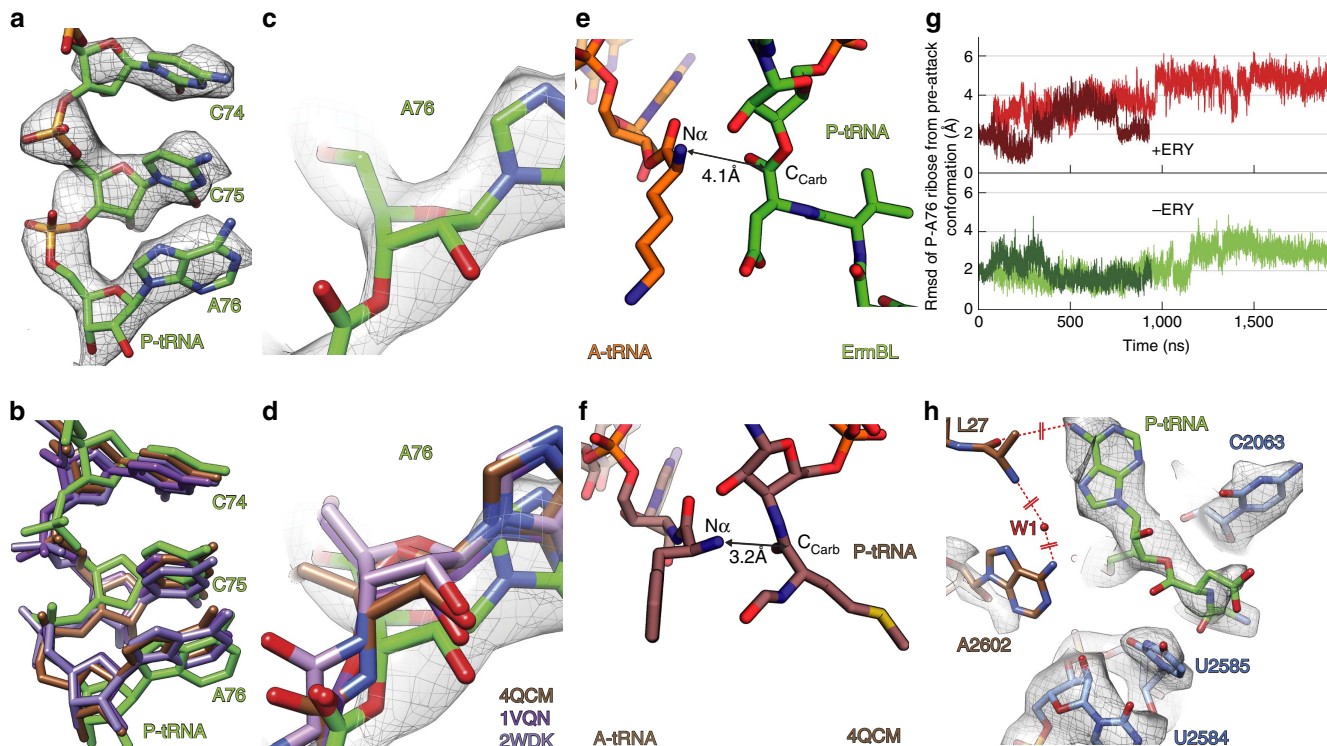

**Figure 6 | Perturbation of the P-tRNA by ErmBL prevents peptide-bond formation.** (**a–d**) Atomic model for the CCA end of the P-site tRNA (green) and (**a,c,d**) electron density (grey mesh) in the ErmBL-APE-SRC compared with (**b**) P-tRNAs in the pre-peptide-bond formation state (brown; PDB4QCM[32]; blue, PDB2WDK[33]), with (**c,d**) zoom onto the ribose of A76. (**e,f**) Relative orientation of the A-site α-amino group and the P-site carbonyl carbon in (**e**) ErmBL-APE-SRC and (**f**) the pre-catalysis state[32]. (**g**) Deviation of the ribose of A76 from the pre-attack state as a function of simulation time. (**h**) Conformation of the PTC in the ErmBL-SRC overlayed with the network of hydrogen bonds, creating a proton wire that enables peptide-bond formation. Cryo-EM map (grey mesh) with models for 23S rRNA (blue), L27 (brown) and P-tRNA (green).

Fig. 8). In the pre-attack complexes, nucleotide A2602 and the N-terminal regions of L27 become ordered[32,33], which, together with the non-bridging phosphate-oxygen (OP1) of A76 of the A-tRNA and the 2′ OH of A2451, coordinate W1 (ref. 32; Fig. 6h and Supplementary Fig. 8). In the ErmBL-SRC, we observed no density for A2602 nor for the N terminus of L27, both of which undergo large fluctuations in the MD simulations, suggesting that the W1 is not firmly coordinated (Fig. 6h). Stabilization of the very N-terminus of L27 has only been observed in the pre-attack structures because of interaction of the side chain of His3 of L27 with the OP1 of C75 of the A-tRNA and the backbone carboxyl of Ala2 of L27 with the N6 of A76 of the P-tRNA[32,33]. Such simultaneous interaction of L27 with the A- and P-tRNA would be prohibited in the ErmBL-SRC by the shifted A76 nucleobase of the P-tRNA (Fig. 6h and Supplementary Fig. 8). We note however that previous studies have demonstrated that ribosomes lacking L27 are not impaired for peptide-bond formation[34] and that mutations of A2602 affect predominantly peptidyl-tRNA hydrolysis rather than peptide-bond formation[35]. Since the position of the ribose 2′-OH of the A76 of the P-tRNA is also critical for the proposed proton shuttles[36–38], it is perhaps more plausible that the perturbed P-tRNA ribose orientation seen in ErmBL plays a more direct role in the impairment of the ribosomal peptidyl-transferase activity.

**ERY affects positioning of the A-site amino group**. During analysis of the MD simulations we noticed that the presence and absence of ERY differentially influenced the conformational dynamics of the neighbouring 23S rRNA nucleotides U2504–U2506. Previous MD simulations on ERY bound to empty ribosomes also observed an allosteric influence of the drug on nucleotides at the PTC, but identified more distant effects on nucleotides U2585 and A2602 (ref. 39). In our study, principal component analysis (see Methods) was performed on the backbone atoms to capture their collective motion, which revealed that, in presence of ERY, the conformation of U2504–U2506 stayed close to that observed in the cryo-EM structure (with an average r.m.s.d. over the simulation of 1.3 Å, red trace in Fig. 7a, see also Supplementary Fig. 9). In contrast, in the simulation performed without ERY (green trace in Fig. 7a), these nucleotides markedly depart from the conformations observed in the simulations with ERY, and, after ~1.5 μs, these nucleotides attained a new conformation (average r.m.s.d. 2.3 Å). Analysis of the structure suggests that it is the desosamine and cladinose sugars of ERY that mediate this conformational change via direct interactions with the backbone and base of G2505 (Fig. 7b). The nucleotides U2504, U2506 together with C2452 comprise the A-site crevice into which the aminoacyl moiety of the A-tRNA is accommodated during peptide-bond formation. In the simulations, the long charged side chain of Lys11, which is sandwiched between the crevice and the P-site, interacts with the crevice nucleotides. In the new conformation adopted in the absence of ERY, the crevice nucleotides are closer to the P-site than in the presence of the antibiotic (Fig. 7b, and distance d1 in Fig. 7c). As a consequence, the A-site Lys11 also moves closer to the P-site. This movement reduces the distance d2 between the α-amino group of the A-tRNA and the carbonyl carbon of the P-tRNA, while in the presence of ERY both the crevice nucleotides (d1) and the Lys11 (d2) are further away from the P-site (Fig. 7b and upper panel in Fig. 7d). We assume that the rate of the chemical step of peptide-bond formation is faster than the rate of reaching conformations where this chemical step is possible. Therefore, simulations that more closely approach these conformations (that is, smallest d2 values) are assumed to represent the complexes with the fastest peptide-bond formation

rates. Importantly, the short d2 distances observed in the pre-attack complexes[32] are approached only in the simulations without the antibiotic (green trace in Fig. 7d), suggesting that ERY can influence the efficiency of peptide-bond formation by restricting the mobility of the A-site crevice and thereby preventing the movement of the aminoacyl moiety of the A-tRNA towards the P-site.

In light of our findings that the motion of the positively charged Lys11 side chain of the A-tRNA is coupled to the negatively charged nucleotides within the A-site crevice, we rationalized that the reduction in stalling efficiency of a Lys11Ala mutation in ErmBL[9] may be because of the loss of interaction between the A-tRNA and the A-site crevice. To test this idea, we extended the MD simulations of the ErmBL-SRC in the presence and absence of ERY by 1 μs following in silico mutation of Lys11 to Ala (Supplementary Fig. 5). In the presence of ERY, the Ala11 moves away from the crevice and towards the P-site (Fig. 7e, lower panel in Fig. 7d and Supplementary Fig. 9), leading to d2 distances shifted towards those seen with Lys11 in the absence of antibiotic (green trace in Fig. 7c). Moreover, the crevice nucleotides retain a similar conformation as in the Lys11 simulation (Fig. 7e, lower panels in Fig. 7a,b and Supplementary Fig. 9), indicating that the motions of Ala11 and the crevice nucleotides are largely uncoupled.

To investigate whether the coupling between the aminoacyl moiety of the A-tRNA and the nucleotides of the A-site crevice is related to the strength of their interaction with one another, we calculated the interaction enthalpies for both Lys11 and Ala11 (see Methods). As predicted, the strongest interactions (more negative interaction enthalpies) were observed for the simulations with Lys11, regardless of the presence or absence of ERY (red and green curves in Fig. 7f, respectively). In contrast, much weaker interaction enthalpies were observed when Lys11 was substituted with Ala11 (cyan and orange curves in Fig. 7f), consistent with the notion that the positively charged lysine side chain is responsible for mediating interaction with the negatively charged nucleotides of the A-site crevice. This finding also suggests that ErmBL-mediated stalling should be influenced by the nature of the A-site aminoacyl moiety and should be efficient when the A-site contains a long positively charged side chain such as arginine. To test this hypothesis, we performed in vitro translation of wild-type ErmBL containing Lys11 in the A-site, or with Lys11 mutated to one of the other 19 proteinogenic amino acids, and used the toe-printing assay to monitor the ribosomes stalled in the presence of ERY (Fig. 7g). Translation was performed in the absence of the amino acid isoleucine (Ile), enabling ribosomes that readthrough the ErmBL stall site to be captured on the subsequent Ile codon of the mRNA, as observed in the absence of ERY (Fig. 7g). Quantitation and adjustment for loading indicated that as expected the nature of the A-site aminoacyl moiety had a dramatic influence on the efficiency of stalling, with Arg11 being threefold more efficient than the wild-type Lys11. It is important to note that for ErmBL comprising A-site amino acids, such as Cys11, Met11 or Phe11, where little stalling was observed, there was also no readthrough to the Ile codon, suggesting that in these cases, ERY induces high levels of peptidyl-tRNA drop-off, as reported previously[11,40,41].

## Discussion
On the basis of our cryo-EM structures and MD simulations of the ErmBL-SRC, we propose a model for how the ErmBL nascent chain, together with the macrolide ERY and components of the ribosome, interplay to inactivate the PTC and induce translational arrest (Fig. 8). In the absence of ERY, and following accommodation of an aminoacyl-tRNA at the A-site of the PTC,

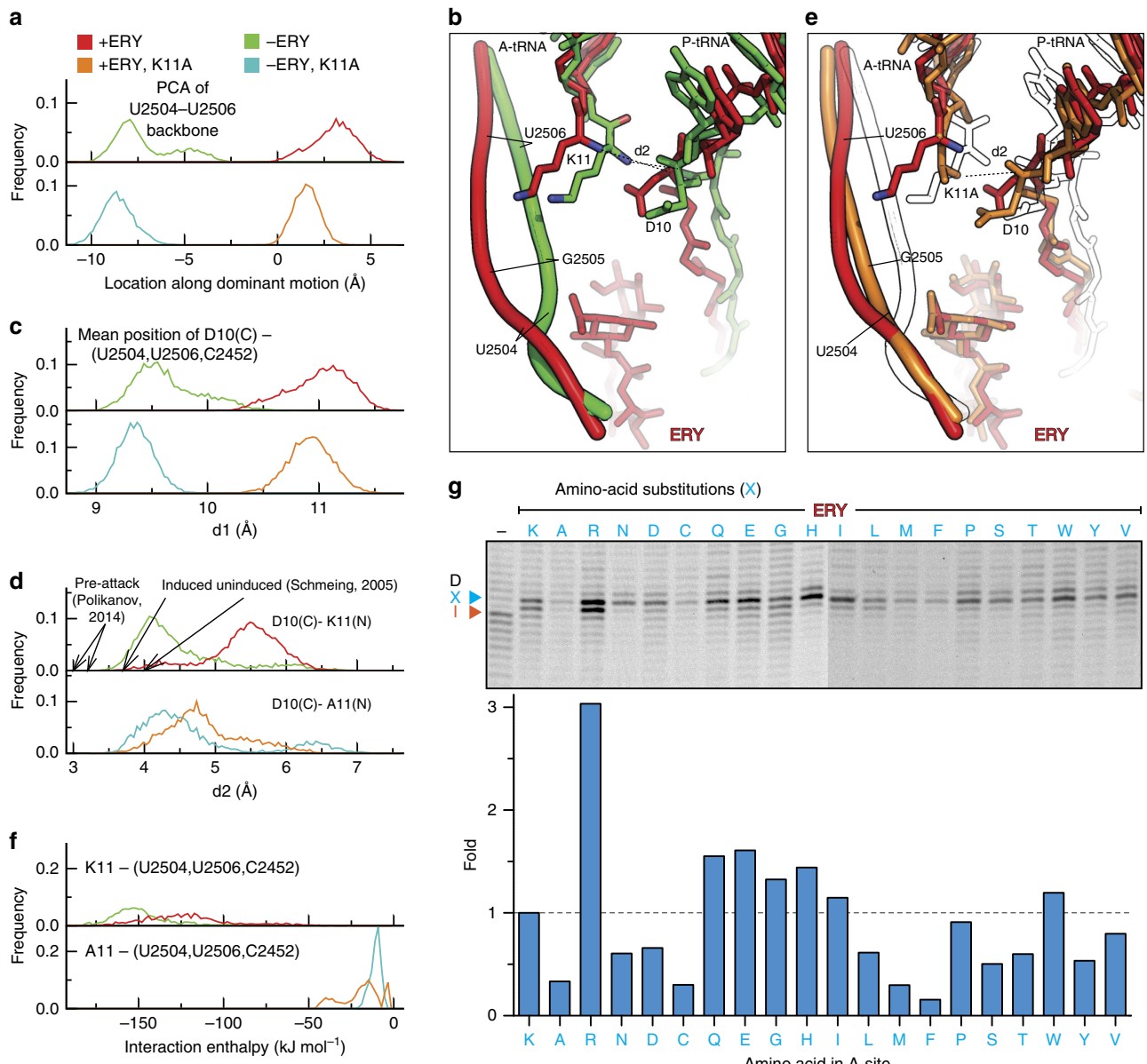

**Figure 7 | Influence of the A-site amino acid and ERY on PTC activity.** (**a**) Movement of the U2504–U2506 backbone in the simulations; histograms of the progression along the most dominant motion for each simulation. WT simulations (upper panel): with ERY (red); without ERY (green). K11A mutation simulations (lower panel): with ERY (orange); without ERY (cyan). (**b**) Comparison of two ErmBL simulation states in presence (red) or absence (green) of ERY. (**c**) Histograms of the distance d1 between the mean position of D10 and crevice nucleotides U2504, U2506 and C2452. (**d**) Histograms of distance d2 between the attacking amide group and the D10 carbonyl carbon. (**e**) Comparison of two simulation states of wild-type ErmBL (K11; red) with the ErmBL K11A mutant (orange) in presence of ERY. The simulation state of wild-type ErmBL in absence of ERY (green in **b**) is superimposed (white). (**f**) Histograms of the interaction enthalpy between the A-site K11 (upper panel) or A11 (lower panel) and the crevice nucleotides U2504, U2506 and C2452.
(**g**) Toe-printing assay to monitor translation of ErmBL variants in presence or absence ( − ) of ERY. Each template contains a Lys substitution in the ErmBL motif. The A-site K of the wild-type motif RNVDK was changed to all other 19 sense amino acids (X). Arrows indicate positions of ribosome accumulation with the indicated substitution/amino acid in the A-site. The blue arrow points to ribosome accumulation at the Asp of the ErmBL RNVDK motif. The orange arrow indicates the position of ribosome accumulation due to Ile depletion. The blue bars represent quantifications of ribosome stalling of the A-site mutants relative to the wild-type ErmBL motif containing Lys. The uncropped gel images are depicted in Supplementary Fig. 10.

the α-amino group of the A-tRNA is positioned to make a nucleophilic attack on the carbonyl carbon of the peptidyl-tRNA in the P-site (Fig. 8a). In all structures to date, the side chain of the C-terminal amino acid, which is attached to the P-tRNA, is oriented such that all nascent polypeptide chains follow a similar path through the upper region of the ribosomal tunnel. This canonical pathway, which we term Path1, is sterically occluded by the presence of macrolide antibiotics (binding site dashed in

Fig. 8a), explaining why this class of antibiotics induces peptidyl-tRNA drop-off for most nascent polypeptide chains[40,41]. In the presence of ERY, we observed a non-canonical path for the ErmBL nascent chain in the cryo-EM structure, which we termed Path2 (Fig. 8b). In the MD simulations, we observe that it is the presence of ERY that restricts the ErmBL conformations towards Path2. These restricted conformations in turn promote interaction of critical residues, such as Arg7 (R7) of ErmBL, with

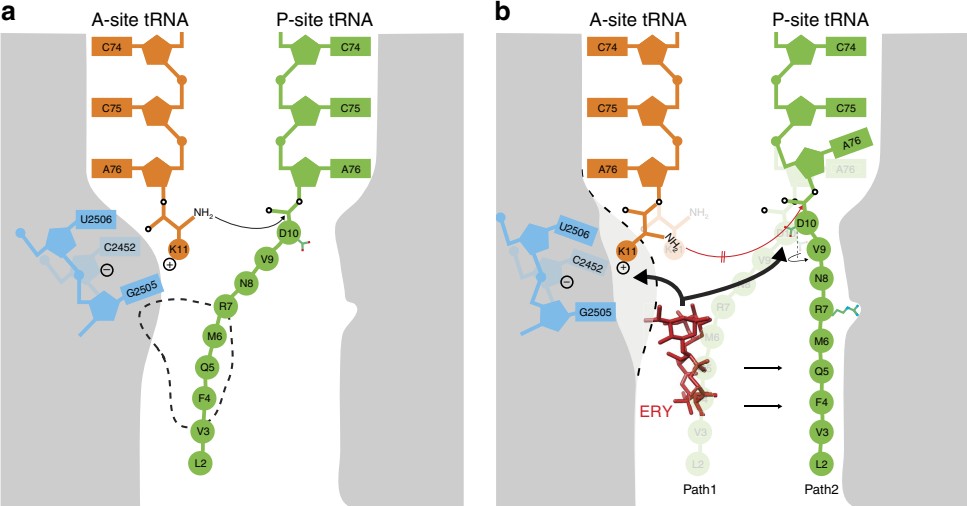

**Figure 8 | Model for ErmBL-mediated translation arrest.** (**a**) Canonical peptide-bond formation in absence of ERY. Here the path of the peptide (Path1) overlaps with the binding site of the drug (broken line). (**b**) Stalling in the presence of ERY. The drug restrains the conformation of the ErmBL peptide (Path2). Relative to **a**, the C-terminal Asp10 side chain is in a rotated position and the P-site tRNA A76 ribose is displaced. The ERY perturbs the A-site crevice nucleotides (blue), which through electrostatic interactions keeps Lys11 away from the P-site. We propose that the increased distance between the attacking amide group of the Lys11 and the carbonyl carbon of Asp10 hinders peptide-bond formation.

23S rRNA nucleotides on one side of the ribosomal tunnel (Fig. 8b). In addition, we observe that Asp10 (D10) of ErmBL adopts an unusual rotated conformation, which together with the R7 interaction, stabilizes the ErmBL nascent chain in the alternative Path2 conformation, rather than the canonical Path1. Path2 is directed away from the tunnel-bound drug, thus preventing direct peptide–drug interactions and explaining why ErmBL stalling occurs with both macrolides and ketolides[9]. Our findings are consistent with a recent study demonstrating that mutations of R7 and D10 not only have a dramatic effect on ErmBL stalling but can also alter the antibiotic specificity, for example, D10G/E or D10P/Y mutations restrict stalling to ERY or Tel, respectively[11]. In addition, it was possible to change the ERY dependency of ErmCL to also include Tel by substituting the Ile9/Ser10 with Asp/Lys combination as in ErmBL[11].

Presumably as a consequence of the unusual rotation of D10 and conformation of the ErmBL nascent chain, we observe that the A76 ribose of the P-tRNA is also perturbed. The perturbed A76 of the P-tRNA has a suboptimal geometry for nucleophilic attack of the α-amino group of the A-tRNA (Fig. 8b), providing a structural explanation for the reduced rate of peptide-bond formation and increased ribosome stalling during translation of ErmBL[9]. In addition, we have observed biochemically and inferred from the MD simulations that the nature of the A-site amino acid also influences the efficiency of translational stalling. On the basis of the MD simulations, we propose that the movement of the A-site crevice nucleotides (G2505-U2506/C2452) is coupled with the movement of the A-tRNA towards the P-tRNA during peptide-bond formation, thus allowing the α-amino group of the A-tRNA to make a nucleophilic attack on the carbonyl carbon of the peptidyl-tRNA in the P-site (Fig. 8a). We propose that the coupled movement results because of the interaction between the positively charged Lys11 (K11) of the A-tRNA and the negatively charged environment of the A-site crevice (Fig. 8a). In the MD simulations in presence of ERY, we observe that the drug interacts and stabilizes a conformation of the A-site crevice that prevents the coupled movement of the A-tRNA towards the P-site, thus suggesting that ERY can influence peptide-bond formation by allosterically influencing nucleotides within the A-site of the PTC (Fig. 8b). Consistently,

uncoupling of the A-tRNA and A-site crevice is observed in the MD simulations with Lys11Ala mutant, indicating that the reduced interaction between Ala11 of the A-tRNA with the A-site crevice allows the A-tRNA to move towards the P-tRNA and undergo peptide-bond formation. Biochemically, we also observed that stable ErmBL-SRC formation is abolished by a Lys11Ala mutation, consistent with previous findings[9], but that also other positively charged amino acids, such as Trp, His, Gln and especially Arg, promoted ribosome stalling (Fig. 7g).

In conclusion, this study illustrates how ERY exerts a dual effect on the PTC to inhibit peptide-bond formation, namely, by (i) restricting the path of the ErmBL such that orientation of the P-tRNA is perturbed and (ii) stabilizing the A-site crevice to prevent movement of the A-tRNA towards the P-site. Inhibition by P-tRNA perturbation has been suggested previously for the SecM arrest peptide[16]. Recent cryo-EM structures of other arrest peptides, including the drug-dependent ErmCL-SRC[13], reveal that PTC silencing is induced by distinct allosteric conformational rearrangements that differ to those reported here for ErmBL. Thus, further investigations will be necessary to ascertain the extent to which other drug-dependent stalling systems utilize similar mechanisms to induce translational arrest or whether further novel mechanisms will be uncovered.

## Methods
**Generation and purification of ErmBL-SRC.** ErmBL-SRCs were generated following the same procedure as previously described[9]. The *2XermBL* construct was synthesized (Eurofins, Martinsried, Germany) such that it contained a T7 promoter followed by a strong RBS spaced by seven nucleotides (nts) to the ATG start codon of the first *ermBL* cistron. A linker of 22 nts separated the stop codon of the first *ermBL* cistron and the start codon of the second *ermBL* cistron. The linker also comprised the strong RBS 7 nts upstream of the ATG start codon of the second *ermBL* cistron, enabling initiation of translation independent from the first *ermBL* cistron. Each *ermBL* cistron encoded amino acids 1–17 corresponding to ErmBL leader peptide (Genbank accession number K00551) present on macrolide resistance plasmid pAM77 from *S. sanguis* strain a1 (ref. 42). The complete sequence of *2XermBL* construct is:

5′-*TAATACGACTCACTATAGGG*AGTTTTTATA**AGGAGG**AAAAAATATG TTGGTATTCCAAAT GCGTAATGTA**GAT**AAAACATCTACTATTTTGAAA TAAAGTTTTTATA**AGGAGG**AAAAAATATGTTGGTATTCCAAATGCGT AATGTA**GAT**AAAACATCTACTATTTTGAAATAA-3′ (T7 Promoter, italics; RBS, bold; ErmBL open reading frame (ORF), shaded grey with GAT codon in the P-site of stalled ribosome shown in bold; annealing site for complementary DNA

oligonucleotide, underlined). *In vitro* translation of the 2XermBL construct was performed using the Rapid Translation System RTS 100 *E. coli* HY Kit (5PRIME; Cat. No. 2401110). Translation reactions were analysed on sucrose density gradients (10–55% sucrose in a buffer A, containing 50 mM HEPES-KOH, pH 7.4, 100 mM KOAc, 25 mM Mg(OAc)$_2$, 6 mM β-mercaptoethanol, 10 μM ERY and 1 × Complete EDTA-free Protease Inhibitor cocktail (Roche)) by centrifugation at 154,693*g* (SW-40 Ti, Beckman Coulter) for 3 h at 4 °C. For ErmBL-SRC purification, disome fractions were collected using a Gradient Station (Biocomp) with an Econo UV Monitor (Bio-Rad) and a FC203B Fraction Collector (Gilson). Purified ErmBL-SRC disomes were concentrated by centrifugation through Amicon Ultra-0.5-ml Centrifugal Filters (Merck-Millipore) according to the manufacturer's protocol. To obtain monosomes of the ErmBL-SRC, a short DNA oligonucleotide (5′-ttcctccttataaaact-3′, Metabion) was annealed to the linker between the *ermBL* cistrons of the disomes, generating a DNA–RNA hybrid that could be cleaved by RNase H (NEB) treatment in buffer A at 25 °C for 1 h. After cleavage of the disomes, ErmBL-SRC monosomes were again purified and concentrated by centrifugation through Amicon Ultra-0.5-ml Centrifugal Filters (Merck-Millipore) according to the manufacturer's protocol.

**Toe-printing assay.** The position of the ribosome on the mRNA was monitored with a toe-printing assay based on an *in vitro*-coupled transcription–translation system with the PURExpress *in vitro* Protein Synthesis Kit (NEB)[43,44]. Briefly, each translation reaction consisted of 1 μl solution A, 0.5 μl Δisoleucine amino-acid mixture, 1.5 μl tRNA mixture, 0.5 μl ErmBL variant solution B, 1 μl (0.7 pmol) DNA template (ermBL variants): (5′-attaatacgactcactatagggatataaggaggaaaacat**atg**ttggtattcca aatgcgtaatgtgagatAAA**ata**tctactattttgaaa**taa**gtgatagaattctatc*gttaataagcaaaattcattat aacc*-3′, with start codon ATG, catch isoleucine codon ATA and stop codon TAA in bold, Lys codon (AAA) in capital letters, the ermBL ORF underlined and primer-binding sites in italics) and 0.5 μl additional agents (nuclease-free water, ERY or Ths). All templates were synthetized via PCR using overlapping forward and reverse primers containing the designated mutation in the Lys position of the ermBL motif. Translation was performed in the absence of isoleucine at 37 °C for 15 min at 500 r.p.m. in 1.5-ml reaction tubes. Ile-tRNA aminoacylation was further prevented by the use of the Ile-tRNA synthetase inhibitor mupirocin (50 μM). After translation, 2 pmol Alexa647-labelled NV-1 toe-print primer (5′-GGTTATAATGAATTTTGCTTATTAAC-3′) was added to each reaction and incubated at 37 °C without shaking for 5 min. Reverse transcription was performed with 0.5 μl of AMV RT (NEB), 0.1 μl dNTP mix (10 mM) and 0.4 μl Pure System Buffer, and was incubated at 37 °C for 20 min. Reverse transcription was quenched and RNA degraded by addition of 1 μl 10 M NaOH and by incubation for at least 15 min at 37 °C, and then was neutralized with 0.82 μl of 12 M HCl. Toe-print resuspension buffer (20 μl) and PN1 (200 μl) buffer were added to each reaction before treatment with a QIAquick Nucleotide Removal Kit (Qiagen). The Alexa647-labelled DNA was then eluted from the QIAquick columns with 80 μl of nuclease-free water. A vacuum concentrator was used to vapourize the solvent, and the Alexa647-labelled DNA was then dissolved into 3.5 μl of formamide dye. The samples were heated to 95 °C for 5 min before being applied on a 6% polyacrylamide (19:1) sequencing gel containing 7 M urea. Gel electrophoresis was performed at 40 W and 2,000 V for 2 h. The GE Typhoon FLA9500 imaging system was subsequently used to scan the polyacrylamide gel. Toe-print intensities were determined using Image Studio Lite Version 5.2 and the uncropped gel images are depicted in Supplementary Fig. 10.

**Negative-stain electron microscopy.** Ribosomal particles were diluted in buffer A to final concentrations of 0.5 A$_{260}$ per ml up to 5 A$_{260}$ per ml in order to determine the optimal ribosome density for cryo-EM. One drop of each sample was deposited on a carbon-coated grid. After 30 s, grids were washed with distilled water and then stained with three drops of 2% aqueous uranyl acetate for 15 s. The remaining liquid was removed by touching the grid with filter paper. Micrographs were taken using a Morgagni transmission electron microscope (TEM; FEI, Eindhoven, the Netherlands), 80 kV, wide angle 1 K charge-coupled device at direct magnifications of 72 K.

**Cryo-EM and single-particle reconstruction.** Four A$_{260}$ per ml monosomes of the ErmBL-SRC was applied to 2-nm pre-coated Quantifoil R3/3 holey carbon-supported grids and vitrified using a Vitrobot Mark IV (FEI). Data collection was performed at NeCEN (Leiden, the Netherlands) on a Titan Krios TEM (FEI) equipped with a Falcon II direct electron detector at 300 kV with a magnification of × 125,085, a pixel size of 1.108 Å and a defocus range of 0.7–1.2 μm. The data are provided as a series of seven frames (dose per frame of 4 e$^−$/Å$^2$) from which we summed frames 2–5 (accumulated dose of 28 e$^−$/Å$^2$) after alignment using the Motion Correction software[45]. Images were processed using a frequency-limited refinement protocol that helps prevent overfitting[46], specifically by truncation of high frequencies (in this case at 8 Å). We did not refine two half data sets independently for resolution determination; nevertheless, as reported and expected[46], we find that using this processing regime the 0.143 FSC value provides a good indicator for the true average resolution of the map. In addition, the local resolution of the map was calculated using ResMap[47]. Power spectra, defocus values, astigmatism and estimation of micrograph resolution were

determined using the CTFFIND4 software[48]. Micrographs showing Thon rings beyond 3.6 Å resolution were further manually inspected for good areas and power-spectra quality. Data were processed further using the SPIDER software package[49], in combination with an automated workflow as described previously[50]. After initial, automated particle selection based on the programme SIGNATURE[51], initial alignment was performed on 285,462 particles, using *E. coli* 70S ribosome as a reference structure[9]. After removal of noisy particles (30,093 particles; 11%), the data set of 255,369 particles could be sorted into two main subpopulations using an incremental K-means-like method of unsupervised three-dimensional sorting (Supplementary Fig. 1)[52]. A minor subpopulation (94,139 particles, 37%) was obtained containing stoichiometric density for the P-tRNA, lacking E-tRNA (L1 stalk in the 'out' position) and containing sub-stoichiometric density for the A-tRNA, which was not sorted further because of the low particle numbers. The major subpopulation (161,231 particles; 63%) was defined by the presence of stoichiometric densities for P- and E-tRNAs and sub-stoichiometric density for the A-tRNA. This population was further sorted into two additional subpopulations, both containing stoichiometric densities for the P- and E-tRNAs (L1 stalk in the 'in' position) and differing by the presence (termed ErmBL-APE-SRC) or absence of A-tRNA (ErmBL-PE-SRC). Both subpopulations could be refined to an average resolution of 3.6 Å (0.143 FSC) and a local resolution extending towards 3.0 Å for the core of the 30S and 50S subunits as computed using ResMap[47] (Supplementary Fig. 2). The final maps were subjected to the programme EM-BFACTOR[53] in order to apply an automatically determined negative B-factor for sharpening of the map.

**Molecular modelling and PDB alignments.** Initial models for the two ErmBL-SRCs were obtained by combining: (i) a model of the *E. coli* 70S ribosome derived from a crystal structure at 2.8 Å resolution (PDB4U27 (ref. 54); (ii) (A)/P/E-tRNAs and mRNA extracted from the 2.55-Å crystal structure of a pre-attack complex of the *T. thermophilus* 70S ribosome (PDB4QCM[32], mutated and remodelled in *Coot*[55] to yield fully modified *E. coli* tRNA$^{Lys}$, tRNA$^{Asp}$ and tRNA$^{Val}$ and a short mRNA, respectively; and (iii) the ErmBL peptide, which was modelled in *Coot*. The ErmBL-APE-SRC and ErmBL-PE-SRC models assembled in this manner were placed into their corresponding density maps using multifragment rigid body refinement in *Situs*[56]. The backbones for the rRNA and tRNA molecules were then fixed using the *Phenix-Erraser* pipeline[57], and atomic coordinates were refined using *phenix.real_space_refine*[58], with base pairs as additional restraints. Model validation was carried out on the MolProbity server[59], and the final model statistics are presented in Supplementary Table 1. Alignment of other ribosome structures to the ErmBL-SRC was performed in Chimera on the basis of the 23S rRNA. The r.m.s.d. between the 23S rRNA of the pre-attack complex (PDB4QCM)[32] and the 23S rRNA of the ErmBL-SRC was 1.1 Å for all atoms and 0.9 Å for all atoms within 30 Å of the ErmBL nascent chain. Similarly, the r.m.s.d. between the 23S rRNA of the accommodated and unaccommodated states (PDB1VQN/1VQ6)[29–31] and the 23S rRNA of the ErmBL-SRC was 2.3 Å for all atoms and 0.86 Å for all atoms within 30 Å of the ErmBL nascent chain.

**Forcefield parameters.** In all MD simulations the amber99sb force field[26] and the SPC/E water model[27] were used. In addition, for K$^+$ and Cl$^−$ ions parameters were taken from ref. 60 and for modified nucleotides parameters were taken from ref. 61. The set of initial coordinates of ERY from the cryo-EM model was protonated and energy-minimized by a HF/6-31G* optimization in GAUSSIAN 09 (ref. 62). The electrostatic potential, calculated from the optimized structure at more than 140,000 points of a molecular surface around the ERY, was fitted to partial charges placed at the atomic positions using the ESPGEN module in the AMBER 11 (ref. 63). All additional force field parameters for ERY were obtained using the ANTECHAMBER module in AMBER. Parameters were converted for use in GROMACS 5.0 using ACPYPE[64]. Esther bonds between amino acids and the A76 of the A- and P-site tRNAs were treated as described earlier[23]. Point charges and atom types for the backbone atoms of the uncharged amino acids lysine and alanine attached to the A-site tRNA were derived as described above for ERY.

**System set-up.** In the initial cryo-EM structure, for the A-site tRNA$^{Lys}$, nucleotides 16–17 were not modelled. To add the missing nucleotides, first, coordinates of nucleotides 15–18 were extracted from the tRNA$^{Phe}$ in a ribosome EF–Tu complex structure[65]. Next, the extracted nucleotides were rigid-body-fitted to the cryo-EM structure using backbone atoms (P, C4′ and O3′) of nucleotides 15 and 18. The fitted nucleotide C16 was mutated to a H2U and then nucleotides 16–17 were included in the model.

In the cryo-EM structure, the tRNA$^{Val}$ in the E-site also lacked coordinates for nucleotides 16–17. These were extracted from tRNA$^{Phe}$ (ref. 65) and rigid-body-fitted in the same manner as described for the A-site tRNA. Here, to match the tRNA$^{Val}$, nucleotides 16 and 17 were mutated to C and 5MU, respectively. Further, the E-site tRNA lacked coordinates for the CCA-tail nucleotides. To model these nucleotides, first, we extracted nucleotides 1 and 72–76 of the E-site tRNA$^{fMet}$ from the ribosome–EF–Tu complex structure[65]. The extracted nucleotides were then rigid-body-fitted to the E-site tRNA in the cryo-EM structure using backbone atoms (P, C4′ and O3′) of nucleotides 1 and 72–73. The fitted nucleotides 74–76 were then included in the model.

The N-terminal end of protein L27 (amino acids 1–10) was not resolved in the cryo-EM structure. To test whether the conformation of L27 in the pre-attack complex is stable in the presence of ERY, the N terminus was modelled on the basis of the crystal structure of a pre-attack state (*T. thermophilus*, PDB ID 1VY4 (ref. 32)). To that aim, first, the phosphates of rRNA nucleotides within a distance of 5 Å of L27 amino acids 1–10 were extracted from the *T. thermophilus* structure. L27 amino acids 1–10 were aligned to the cryo-EM structure by rigid-body fitting to the corresponding phosphate atoms. Finally, to match the *E. coli* sequence[65], amino acids G6 and L7 were mutated to A and G, respectively.

The L1 protein and 23S rRNA H38 nucleotides 886–891 were not modelled in the cryo-EM structure. To model the L1 protein, coordinates of the L1 stalk rRNA (nucleotides 2093–2196) and of the tip of H68 (nucleotides 1860–1883) were extracted from the cryo-EM structure and from all trajectories from previous MD simulations of translocation intermediates that include the L1 protein[24]. The coordinates from each frame of the trajectories were compared with those extracted from the cryo-EM structure by calculating the r.m.s.d. From the frame with the lowest r.m.s.d. (0.38 nm), the coordinates of the L1 protein were extracted and added to our model. To model the missing H38 nucleotides, first, nucleotides 883–885 and 892–893 as well as all residues within a radius of 1.2 nm were extracted from the cryo-EM structure and all trajectories. As described above, from the trajectories, the frame with the lowest r.m.s.d. (0.12 nm) was selected and the coordinates of nucleotides 886–891 were added to our model.

**MD simulations.** Starting structures of three states were used for the MD simulations: the model of the ribosome in complex with tRNAs, ErmBL and ERY (+ERY); the model of the same complex with an alternative conformation of ErmBL (+ERY'); and the model after removal of ERY (−ERY). The alternative conformation of ErmBL (+ERY') was obtained by perturbing the ψ dihedral of ErmBL amino acid Asp10 from 270° to 90°, as described in the following subsection. For each of the three starting structures, two independent simulations were carried out.

The protonation states of amino acids were determined using WHATIF[66]. Each structure was first solvated in a dodecahedron box keeping a minimum distance of 1.5 nm between the model atoms and the box boundaries. Next, the simulation system was neutralized with $K^+$ ions and explicit salt (7 mM $MgCl_2$ and 150 mM KCl) was added using the programme GENION from the GROMACS suite[25].

All MD simulations were carried out with GROMACS 5 (ref. 25). Lennard–Jones and short-range electrostatic interactions were calculated within a distance of 1 nm. Long-range electrostatic interactions, beyond 1 nm, were calculated by particle-mesh Ewald summation[67] with a 0.12-nm grid spacing. The bond lengths were constrained using the LINCS algorithm[68]. The temperature of solute and solvent was controlled independently at $T = 300$ K using velocity rescaling[69] with a coupling time constant of $\tau_T = 0.1$ ps. An integration time step of 4 fs was used, applying virtual site constraints[70]. The coordinates were recorded for analysis every 5 ps.

The system was then equilibrated in four steps:

- Energy minimization using steepest decent.
- 0–50 ns: Berendsen barostat[71] ($\tau_p = 1$ ps) and position restraints on all the heavy atoms included in the cryo-EM structure ($k = 1,000$ kJ $mol^{-1}$ $nm^{-2}$).
- 50–70 ns: Linearly decreasing of the position restraint force constant $k$ to zero.
- 70–2,070 ns: Parrinello–Rahman barostat[72] ($\tau_p = 1$ ps) and no position restraints.

To study the effects of the ErmBL Lys11Ala mutation, the final structure of one trajectory for each state ($t = 2,070$ ns) was extracted. In this structure, the Lys11 of ErmBL was then mutated to an alanine. To preserve the neutral charge of the system, one randomly chosen water molecule, which was at least 2 nm away from any ribosomal atoms, was replaced by a $K^+$ ion. The simulation system was subsequently, first, energy-minimized and, second, equilibrated with Berendsen barostat (as described above) for 20 ns (2,070–2,090 ns) and, finally, with Parrinello–Rahman barostat for 1,980 ns (2,090–3,070 ns).

**Structural deviations.** To monitor the stability of the simulations, the r.m.s.d. of all ribosome–ErmBL–tRNA complex atoms from their initial coordinates (cryo-EM structure) was calculated for each frame after rigid-body fitting of all $C^\alpha$ and P atoms (Supplementary Fig. 5).

To investigate whether, upon removal of ERY, the P-site A76 ribose moves towards its unstalled pre-attack state conformation, for each frame and simulation, the coordinates of the ribose and phosphate group atom of nulceotides C74–C75 of the P-site tRNA were rigid-body-fitted to their coordinates in the pre-attack structure[32]. Finally, the r.m.s.d. of the A76 ribose atoms from their pre-attack conformation was calculated (Fig. 6g in the main text).

**ErmBL peptide dihedral angles.** To monitor the conformational dynamics of the ErmBL peptide, dihedral angles for each amino acid were calculated (Supplementary Fig. 7). In particular, the ψ dihedral of Asp10 was found to spontaneously switch between the two starting conformations (+ERY and +ERY'). Therefore, we consider the simulations with ERY as exploring the same

state. The results from the +ERY' simulations support all the conclusions drawn from the +ERY simulations described in the main text. For clarity, these results are only reported as Supplementary Information.

**Overlap of ErmBL with the volume occupied by ERY.** To answer the question whether ERY restricts the conformation of the ErmBL peptide for each simulation and all frames at intervals of 1 ns, the overlapping volume between the peptide coordinates and the coordinates of ERY from the cryo-EM structure were calculated after alignment of the surrounding 23S rRNA phosphates. To that aim, the vander-Waals volume of ERY alone in the starting structure $V_e(0)$, of the peptide alone $V_p(t)$ and of ERY and peptide together $V_{ep}(t)$ were calculated for each time $t$ using the GROMACS programme gmx sasa[25]. The overlapping volume was calculated as $V_o(t) = V_e(0) + V_p(t) − V_{ep}(t)$, see Fig. 2f in the main text.

**Conformation of 23S nucleotides 2504–2506.** In the simulations, 23S rRNA nucleotides 2504–2506 were found to change conformation upon removal of ERY. To quantify the conformational change, a principal component analysis[73] was carried out. First, we identified all P atoms of the 23S nucleotides located within a distance of 8 Å from the CCA tails of A- and P-site tRNAs and ErmBL amino acids 2–11 in the cryo-EM structure. Second, all trajectories were superimposed by least-square fitting using these P atoms. Then, the coordinates of the backbone atoms of nucleotides 2504–2506 were extracted from the superimposed trajectories, concatenated and the atomic displacement covariance matrix was calculated. Finally, the trajectory of each simulation was projected on the first eigenvector of this matrix (Fig. 7a and Supplementary Fig. 9).

**Distance of the crevice nucleotides from the A-site.** The positively charged side chain of the ErmBL amino acid Lys11, which is attached to the A-site tRNA in the stalled complex, was found to interact with 23S nucleotides U2504, U2506 and C2452. The side chain specifically interacts with certain parts of these nucleotides: the U2504 ribose atoms, U2506 phosphate atoms and the C2452 base atoms. To measure the width of the A-site crevice[74], first, the centres of mass of these nucleotide parts were calculated for each frame in each of the superimposed trajectories (see section Conformation of 23S nucleotides 2504–2506). Next, the mean position of the carbonyl atom of the ErmBL amino acid Asp10, which is attached to the P-site tRNA, was calculated from all superimposed trajectories. Finally, to quantify the width of the A-site crevice, the distance between this mean position of the carbonyl atom and centre of geometry of the three centres of mass (U2504, U2506 and C2452) was calculated (d1 in Fig. 7c and Supplementary Fig. 9).

**Interaction enthalpies of Lys11 and Ala11.** To estimate the relative strength of the interaction of the A-site tRNA amino acids Lys11 and Ala11 with 23S nucleotides U2504, U2506 and C2452, the interaction enthalpies, that is, the sum of electrostatic and vander-Waals interactions were calculated. The electrostatic and vander-Waals interactions were calculated with the point charges and Lennard–Jones parameters from the amber99sb forcefield[26] using GROMACS[25]. The sum of these interaction enthalpies is shown in Fig. 7f and Supplementary Fig. 9. Note that the estimated interaction strength calculated here represents a qualitative estimate of the free energy differences and does not include entropic contributions.

**Hydrogen bonds.** Hydrogen bonds of the A76 of A-site and P-site tRNAs and ErmBL amino acids 6–11 were monitored for all simulations using the GROMACS programme gmx hbonds[25]. For all hydrogen bonds with an occupancy >20% in at least one simulation, the hydrogen bond occupancy was averaged over windows of 20 ns (Supplementary Fig. 6).

**Data availability.** The cryo-EM maps and associated atomic coordinates have been deposited in the EMDB and PDB with the accession codes EMDB-8175 and PDB ID 5JTE (ErmBL-APE-SRC) and EMDB-8176 and PDB ID 5JU8 (ErmBL-AP-SRC). The additional MD data that support the findings of this study are available from the corresponding author upon request.

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

## Acknowledgements

We thank Rishi Matadeen and Sacha DeCarlo for data collection at the NeCEN facility (Leiden, the Netherlands) and Charlotte Ungewickell for expert technical assistance. We thank the computer centre Garching (RZG) and the Gesellschaft für wissenschaftliche Datenverarbeitung Göttingen (GWDG) for technical assistance; computer time has been provided by the RZG. This research was supported by the Deutsche Forschungsgemeinschaft (FOR1805 to D.N.W., A.C.V. and L.V.B; WI3285/3-1 and GRK1721 to D.N.W.) and the Agence Nationale de la Recherche (ANR-14-CE09-0001 to C.A.I. and D.N.W.).

## Author contributions

S.A. prepared ErmBL-SRC and performed single-particle cryo-EM analysis. S.A. and C.A.I modelled and refined the ErmBL-SRC. L.V.B., H.G. and A.C.V. performed MD simulations. M.G. performed toe-printing analysis. All authors analysed data, prepared figures and wrote the paper.

## Additional information

**Competing financial interests:** The authors declare no competing financial interests.

