## [Peer review file · Nature Communications]

Reviewers' Comments:

Reviewer #1 (Remarks to the Author)

The authors combined cryo-EM and MD simulation to study the molecular mechanism of ErmBL-mediated ribosome stalling. Overall, this is a nice piece of example showing the how atomic mechanisms could be derived by a combination of static high-resolution structure and dynamic information from in silico simulation.

Major Issues:

1. It would be appropriate to show the density and fitting of Erm in the context of the cryo-EM map.
2. Throughout figures, most of the data are presented by comparison of ErmBL-SNC with reference structures. Ribosome is a gigantic structure with many individual components. It would be essential to indicate how the structural alignments were performed. These information could be provided in Figure legends or methods.
3. On page 11, the authors make a statement that Lys-tRNA movements, compared to unaccommodated and accommodated states, are 0.9 and 0.6 Å, respectively. This is far beyond the nominal resolution of the density map. Also, for the same reason in issue #2, how was the alignment done? What is the rmsd of the alignment? This could help reader understand the significance of reported structural differences. The same question also applied to Fig. 5e (statement in the first paragraph of Page 12)
4. It becomes mandatory to validate the atomic model derived from cryo-EM maps to avoid overfitting. Have the authors done model validation?

Minor issues:

1. Can the authors comment why they did not use Relion to do 3D classification? Experiences from many labs show that Relion gives the best results from the same data sets.
2. The CTF fitting was done with Spider "TF ED". From our experience, CTFFIND3/4 gives better estimation of CTF parameters. Replacing the CTF estimates in the final refinement might give the authors a slight boost on reported resolution.
3. SRC was not defined at its first appearance (page 4)

Reviewer #2 (Remarks to the Author)

The macrolide antibiotic erythromycin is a generic medicine widely used clinically for the treatment of bacterial infections, in particular those of newborn children or in patients with a medical history of allergy to penicillin or its derivatives. In the opportunistic pathogen *Streptococcus sanguinis* resistance to erythromycin is conferred upon translational arrest due to detection and stabilisation of the presence of the macrolide within the ribosomal nascent peptide exit channel by the ErmBL peptide, leading to expression of ErmB.

In this manuscript Arenz and colleagues describe 3.6 Å cryo-EM structures of ErmBL-stalled ribosome complexes in the presence and absence of A-site tRNA and interpret their results with the aid of computational modeling and molecular dynamics simulations. The study represents the continuation of a previous Nature Communications study published in March 2014, "Molecular basis for erythromycin-dependent ribosome stalling during translation of the ErmBL leader peptide, Arenz et al., Nat. commun. 5:3501", at a headline resolution of 4.5 Å. The increase in resolution, while modest, does bring the structure from intermediate to near-atomic resolution, allowing atomic coordinate refinement of the model, and making the molecular dynamics simulations plausible. The authors identify a series of changes in the conformations of both the ErmBL peptide

and the peptide channel that lead them to propose a plausible mechanistic model for stalling. While this model cannot be considered secure without much more biochemical data, the reported incremental advance, will be of interest in the field.

Specific points:

- The authors note that the mutation V9A abolishes stalling, and speculate on this subject within their manuscript, but do not take the obvious step of carrying the mutation into their molecular dynamics simulation in silico (as they do for K11A) and reporting the results. While exhaustive interrogation of the system is probably not feasible, specific mutations that are known to affect the system, or about which the authors seek to draw conclusions, should most certainly be investigated.
- The argument above must also apply to the other mutations highlighted in the study, particularly EMBL R7A, N8A and D10A, and probably the compensatory mutations noted for U2586. The rate at which a position allowing nucleophilic attack is attained could be reported for each mutation, or a similar statistic if preferred. Such data would confirm or deny the utility of the predictions the authors make from their molecular dynamics simulations.
- The authors have not reported the value of any target statistic or unrestrained measure to prevent over fitting. Indicative values should be calculated and reported in the table.
- The authors have used truncation of spatial frequencies at 8 \AA in lieu of independently refined half sets in their final refinements and for resolution determination. They correctly state that this should inhibit over-fitting, however we note that the 0.143 FSC threshold is defined with respect to independent half sets (Rosenthal & Henderson, 2003) and has become synonymous with "gold-standard" refinement. Therefore, the protocol must be described and stated much more clearly throughout. It should not be necessary to dissemble if there are no problems with the outcome; the methodology must be explained directly and clearly to avoid confusing the reader.

Reviewer #3 (Remarks to the Author)

The manuscript "A combined cryo-EM and molecular dynamics approach reveals the mechanism of ErmBL-mediated translation arrest" by Stefan Arenz and co-workers describes a new 3.6A-resolution cryo-EM structure of the ErmBL stalled ribosome complex and subsequent molecular dynamic simulations using the stalled complex. The current structure is significantly improved over a similar structure reported by the Arenz et al in 2014. The authors use this high-resolution structure as constraints for subsequent MD simulations in the presence and the absence of erythromycin. A detailed analysis of the structure and dynamics of the complex is reported and analyzed on the background of the complex after in silico removal of the bound erythromycin. Based the combined structural, biochemical and in silico data the authors present a model where erythromycin has a dual effect both on the positioning of the CCA-end of the peptidyl-tRNA, as well as on the A-site bound aminoacyl-tRNA, preventing efficient peptide bond formation on the ErmBL peptide leading to stalling of translation. The authors in particular highlight the importance of the A-site aminoacyl-tRNA and its amino acid identity on stalling in conjunction with the structural dynamics of the "A-site crevice" (U2506, C2452, G2505) and its modulation by erythromycin binding. The manuscript is well written easy accessible and the findings reported provide important insight into the structural dynamics that interacting small molecular inhibitors can exploit to achieve their effect on the ribosome. The observations reported constitute important contributions to our understanding of erythromycin induced stalling on the ErmBL peptide and as such should be published in Nature Communications after addressing the mostly minor concerns below.

Recently, another report has been published in Nature Chemical Biology (Gupta et al.: Nascent peptide assists the ribosome in recognizing chemically distinct small molecules, DOI: 10.1038/NCHEMBIO.1998, some of the authors were also co-authors on the 2014 report by Arenz et al. in Nature communications reporting a lower resolution complex) also investigating the erythromycin stalled ermBL peptide complex using MD simulations to understand the molecular mechanism of stalling by the antibiotic and ErmBL peptide. The information reported in this report (Gupta et al.) complement the observations in the manuscript under review and should be discussed in the light of the model and data reported.

Minor issues:

P2: Abstract, *Streptococcus sanguis* is not italicized.

P4: The description about compensatory mutations in the 23S rRNA alongside the ErmBL-R7A mutation mention that the U2586 nucleotides compromised stalling efficiency, but then a point is made that the A2062 mutation does not. It is confusing what the relevance of which mutation compensates and which do not at this point in the manuscript without any context to the positions of those nucleotides in the 23S rRNA. Perhaps just mentioning the location and importance of these nucleotides in the introduction might alert the reader to their importance later in the manuscript.

P7 and 8: "The resulting simulations were very stable, with root mean square deviations (rmsd)..." its not clear what is meant by "very stable". The authors should refrain from the use of this qualifier unless they provide a scale that helps assess the claim. e.g. in energy or % present over simulation time.

P8: When describing the flexibility of the N-terminal residues during the course of the simulation it is referred to as "the first four N-terminal residues" which is confusing when this is referring to residues 2-5 that are observable in the cryo-EM structure used for the simulation not the actual first 1-4 amino acids of the ErmBL peptide.

P8: "The absence of the drug... (Fig. 2e,f)" Could 2d not also be cited here?

P12: Paragraph starting "The most recent model for peptide bond formation,..." describes a proton wire model of peptide bond formation yet the description does not match Fig. 6h. Specifically, it is mentioned that water 1 (W1) accepts a proton transferred by A76 of the P-site tRNA and A2451 of the 23S rRNA, but neither of these residues is shown in Fig. 6h.

Also in this paragraph the acronym OP1 is used, followed by O1P later the paragraph, and nowhere is it mentioned what these acronyms are in reference to.

Finally, also in this paragraph, the sentence "Such simultaneous interaction of L27 with A- and P-tRNA..." mentions that in the ErmBL-SRC a simultaneous interaction of L27 to both the A- and P-tRNA is not possible, but the figure referenced to show this (Fig. 6h) does not adequately portray this as the A-site tRNA is not shown. It can be deduced from previous figures and text that the A-tRNA is far enough away that L27 does not make simultaneous interaction with both tRNAs but Fig.6h. does not show this.

Figure 6h - The figure legend and/or description in the text of this figure sub panel does not provide any information about the proton wire model past water 1 (W1), specifically that the hydrogen bonds to W2 and W3 are not known as indicated by the (?) on the figure. The sub panel shows the proton wire between L27 and A2602 as the authors mention in the text. At least some more detail explaining the used symbols would help.

Supplemental Fig. 6 needs a more descriptive figure caption to adequately describe the data being presented in the figure for those less experienced in bioinformatics analyses of MD data. Also highlighting rows of significance that are referenced in the text allows the reader to quickly find

the information.

P13: "conformation of U2504-U2506 stayed close to that observed in the cryo-EM structure": the authors should specify what that means e.g. in RMSD or just distance.

P15: typo: Sentence "To test this hypothesis,..." aminoacids should read amino acids

Reviewer #4 (Remarks to the Author)

In this manuscript, the authors employ a combination of cryo-EM and simulations to elucidate the mechanism of ErmBL-mediated translation arrest. The manuscript is written very well. A strength of the manuscript is that the simulations employed are very long (microseconds). As far as I know, there is only simulation of a full ribosome that has exceeded one microsecond, which highlights the dedicated effort the authors have made to tackling this problem. In summary, the authors use cryo-EM reconstructions to construct atomic models. These models are then used as the starting point for simulations. Together, the authors provide compelling evidence that the primary role of Erm is to impose a steric constriction on the nascent protein, which results in distortion of the CCA end of the PTC. This is proposed as the mechanism of stalling.

There are a few technical points and phrasing clarifications that are necessary before publication.

- 1) A few of the claims are overstated. For example, on page 5 it is stated that "MD simulations reveal that the presence of erythromycin also influences the position of the aminoacyl moiety of the A-tRNA via a cascade of 23S rRNA rearrangements." This is a prediction of the simulations, it is not shown to be true.
- 2) A major claim of this study is that Erm induces a distortion in the CCA end. However, the reported simulations are several microseconds. There should be more complete analysis showing that there isn't some other effect that is causing the distortion. For example, in another microsecond explicit-solvent simulation of the ribosome (Whitford et al PLOS Comp Bio 2013), it was shown that the subunits can rotate several degrees. The author do state that the RMSD is 5.2Å, but this sounds like a large value. What are the rotation angles over that course of time? It is possible that rotations could strain the tRNA molecules, which would also contribute to distortions. Also, where were the largest changes in conformation (large RMSD per residue)? Since there are hundreds of thousands of atoms, an average of 5.2 could still involve extremely large deviations of some regions.
- 3) On page 8, it is claimed that hydrogen bonds between Asp 10 and G2061 were very stable in all simulations. While the authors did perform long simulations, by computational standards, showing that hydrogen bonds are formed for a few microseconds does not mean they are "very stable". This claim should be presented in a quantitative fashion. For example, for what fraction of the simulated time were these hydrogen bonds formed?
- 4) Related to point 3: On page 9, it is stated that hydrogen bonds were observed between C2063 and U2441. Again, a fraction of time formed would be helpful. Figure S6 is not easy to interpret.
- 5) On page 9, it is stated that " mutation of U2586 were able to restore stalling" Since this is referring to a previous publication, it should probably be "is able to restore stalling". As written, it sounds like it is a finding from the current study.
- 6) On pages 11 and 12, there are multiple claims of shifts in positions. Since there is no absolute reference frame for comparison, it is not clear what exactly these values signify. These atoms move relative to what?

7) On page 14, it is stated "Given that peptide bond formation takes place on timescales much shorter than those in the simulations, it is the smallest d_2 values observed in the simulations that must be taken into account" According to biochemical data, peptide bond formation can be the rate limiting step (Molecular Cell 30, 589-598, 2008), and rates of 250,000/s (corresponding to the timescale of the simulation) have not been reported. Please clarify this statement.

8) Page 14: "To test this prediction" The previous statement is not a prediction, it is notion/idea/suggestion/rationalization. Since the current paper has actual predictions, it is confusing to use the term in this context. Following this, it is stated "As predicted, the strongest interactions...." This is very misleading. That is, within in this statement, a hypothesis is treated as a prediction, and a simulation is treated as a test of the prediction. A more appropriate phrasing would be "As expected/suggested, the model predicts that..."

9) Similarly to the previous point, on page 15, it is stated that "much weaker interaction enthalpies were observed when Lys11..." Since this manuscript provides a combined description of experiments and computation, in order to clearly distinguish between theoretical and experimental results, the computational results should be described as predictions and experiments should be reported as observations.

Reviewers' comments:

Reviewer #1 (Remarks to the Author):

The authors combined cryo-EM and MD simulation to study the molecular mechanism of ErmBL-mediated ribosome stalling. Overall, this is a nice piece of example showing the how atomic mechanisms could be derived by a combination of static high-resolution structure and dynamic information from in silico simulation.

Major Issues:

1. It would be appropriate to show the density and fitting of Erm in the context of the cryo-EM map.

We have now included two additional panels (m,n) in the Supplementary Fig 3 showing the density and fitting of the model of the ErmBL-APE-SRC and ErmBL-PE-SRC in the context of respective cryo-EM maps.

2. Throughout figures, most of the data are presented by comparison of ErmBL-SNC with reference structures. Ribosome is a gigantic structure with many individual components. It would be essential to indicate how the structural alignments were performed. These information could be provided in Figure legends or methods.

The structure alignments were performed on the basis of the 23S rRNA and were performed in Chimera. The rmsd of the fitted structures (PDBs 4qcm, 1vqn 1vq6,) to the 23S rRNA of the ErmBL-SRC was 0.9 Å for all atoms within 30 Å of the ErmBL nascent chain. This is now described in the Methods on page 23.

3. On page 11, the authors make a statement that Lys-tRNA movements, compared to unaccommodated and accommodated states, are 0.9 and 0.6 Å, respectively. This is far beyond the nominal resolution of the density map. Also, for the same reason in issue #2, how was the alignment done? What is the rmsd of the alignment? This could help reader understand the significance of reported structural differences. The same question also applied to Fig. 5e (statement in the first paragraph of Page 12)

As mentioned above in point 2, the structure alignments were performed on the basis of the 23S rRNA and were performed in Chimera. This is now mentioned in the Methods, including the rmsd values of the aligned PDBs (1vq6, 4qcm and 1vqn) to the 23S rRNA of the ErmBL-SRC. We agree it can be debated as to whether at 3.6 Å resolution differences of 0.6-0.9 Å can be resolved. However, we do not want to make a claim on the basis of 0.9 Å and 0.6 Å as to whether our model is unaccommodated or accommodated but rather just present the analysis. This is the reason why we also show in Fig. 5g the density for U2506 and the similarity with 1VQN, which suggests that the A-tRNA in the ErmBL-SRC is "fully or near-fully accommodated" as stated on page 12, line 2.

4. It becomes mandatory to validate the atomic model derived from cryo-EM maps to avoid overfitting. Have the authors done model validation?

We have now included a model versus map validation in Supplementary Fig. S2D,H showing that the $FSC_{model\ vs\ map\ 0.5}$ value of 3.7 Å is similar to the $FSC_{half\ maps\ 0.143}$ value of 3.6 Å indicating absence of over-fitting, as discussed by Scheres and Chen (now reference 53).

Minor issues:

1. Can the authors comment why they did not use Relion to do 3D classification? Experiences from many labs show that Relion gives the best results from the same data sets.

In our experience we observe no significant difference in the results from the in silico sorting protocol of Loerke et al that we implement in SPIDER versus the maximum likelihood method utilized by Relion. We note that for the sorting implemented here of the ErmBL-SRC we are separating ribosomal particles containing the presence or absence of A-tRNA and/or E-tRNA (as seen in Supplementary Figure 1), which can be easily validated by looking at the density. An additional reason for not using Relion is because it is computational more demanding (in our estimates >10x slower). The effectiveness of the SPIDER sorting implemented here is moreover validated by the excellent resolution of 3.6 Å, which we believe is sufficient to draw the conclusions that we have made in the manuscript.

2. The CTF fitting was done with Spider "TF ED". From our experience, CTFFIND3/4 gives better estimation of CTF parameters. Replacing the CTF estimates in the final refinement might give the authors a slight boost on reported resolution.

We thank the reviewer for commenting on this since this is an error in the Methods. We had initially used TF ED for CTF estimation but subsequently reprocessed the dataset with CTFFIND4 when the software became available mid way through last year. As the reviewer correctly states we did see a slight boost in resolution of 0.2 Å (from 3.8 Å to 3.6 Å) when using CTFFIND4 compared to TF ED. The correction has now been made in the methods on page 20 including the citation to Rohou et al 2015 (reference 55).

3. SRC was not defined at its first appearance (page 4)

We have now defined SRC in its first appearance on page 4 and removed the definition on page 6.

Reviewer #2 (Remarks to the Author):

The macrolide antibiotic erythromycin is a generic medicine widely used clinically for the treatment of bacterial infections, in particular those of newborn children or in patients with a medical history of allergy to penicillin or its derivatives. In the opportunistic pathogen Streptococcus sanguinis resistance to erythromycin is conferred upon translational arrest due to detection and stabilisation of the presence of the macrolide within the ribosomal nascent peptide exit channel by the ErmBL peptide, leading to expression of ErmB.

In this manuscript Arenz and colleagues describe 3.6 Å cryo-EM structures of ErmBL-stalled ribosome complexes in the presence and absence of A-site tRNA and interpret their results with the aid of computational modeling and molecular dynamics simulations. The study represents the continuation of a previous Nature Communications study published in March 2014, "Molecular basis for erythromycin-dependent ribosome stalling during translation of the ErmBL leader peptide, Arenz et al., Nat. commun. 5:3501", at a headline resolution of 4.5 Å. The increase in resolution, while modest, does bring the structure from intermediate to near-atomic resolution, allowing atomic coordinate refinement of the model, and making the molecular dynamics simulations plausible. The authors identify a series of changes in the conformations of both the ErmBL peptide and the peptide channel that lead them to propose a plausible mechanistic model for stalling. While this model cannot be considered secure without much more biochemical data, the reported incremental advance, will be of interest in the field.

Specific points:

1. The authors note that the mutation V9A abolishes stalling, and speculate on this subject within their manuscript, but do not take the obvious step of carrying the mutation into their molecular dynamics simulation in silico (as they do for K11A) and reporting the results. While exhaustive interrogation of the system is probably not feasible, specific mutations that are known to affect the system, or about which the authors seek to draw conclusions, should most certainly be investigated. The argument above must also apply to the other mutations highlighted in the study, particularly EMBL R7A, N8A and D10A, and probably the compensatory mutations noted for U2586. The rate at which a position allowing nucleophilic attack is attained could be reported for each mutation, or a similar statistic if preferred. Such data would confirm or deny the utility of the predictions the authors make from their molecular dynamics simulations.

We also believe the proposed simulations would be a valuable addition to the existing experimental results on such mutations. It should be noted however that, as also mentioned by reviewer #4, the simulations presented here are extremely computationally expensive. In fact, the simulations presented here are the longest simulations of full ribosomes to date. The suggested mutations would require a total of ~18 months of computation time at a high-performance supercomputing center. We therefore believe that such a large-scale simulation endeavor would go beyond the scope of the present work. We carefully chose the K11A mutation as the most promising to shed light on the A site effects, which could not be inferred from structural or biochemical data alone. Nevertheless, we are considering these as well as other simulations as candidates for future studies aimed at adding pieces to the multifaceted puzzle of controlled ribosomal stalling.

2. The authors have not reported the value of any target statistic or unrestrained measure to prevent over fitting. Indicative values should be calculated and reported in the table.

We have now included a model versus map validation in Supplementary Fig. S2D,H showing that the $FSC_{model\ vs\ map\ 0.5}$ value of 3.7 Å is similar to the $FSC_{half\ maps\ 0.143}$ value of 3.6 Å indicating absence of over-fitting, as discussed by Scheres and Chen (reference 53).

3. The authors have used truncation of spatial frequencies at 8 Å in lieu of independently refined half sets in their final refinements and for resolution determination. They correctly state that this should inhibit over-fitting, however we note that the 0.143 FSC threshold is defined with respect to independent half sets (Rosenthal & Henderson, 2003) and has become synonymous with "gold-standard" refinement. Therefore, the protocol must be described and stated much more clearly throughout. It should not be necessary to dissemble if there are no problems with the outcome; the methodology must be explained directly and clearly to avoid confusing the reader.

We now more clearly state in the methods on page 21 that we indeed did not refine two half datasets independently. For average resolution determination we use the FSC 0.143 value because it accurately reflects the features of the map as seen in Figure 1d-f and agrees with the local resolution calculations based on ResMap.

Reviewer #3 (Remarks to the Author):

The manuscript "A combined cryo-EM and molecular dynamics approach reveals the mechanism of ErmBL-mediated translation arrest" by Stefan Arenz and co-workers describes a new 3.6Å-resolution cryo-EM structure of the ErmBL stalled ribosome complex and subsequent molecular dynamic simulations using the stalled complex. The current structure is significantly improved over a similar structure reported by the Arenz et al in 2014. The authors use this high-resolution structure as constraints for subsequent MD simulations in the presence and the absence of erythromycin. A detailed analysis of the structure and dynamics of the complex is reported and analyzed on the background of the complex after in silico removal of the bound erythromycin. Based the combined structural, biochemical and in silico data the authors present a model where erythromycin has a dual effect both on the positioning of the CCA-end of the peptidyl-tRNA, as well as on the A-site bound aminoacyl-tRNA, preventing efficient peptide bond formation on the ErmBL peptide leading to stalling of translation. The authors in particular highlight the importance of the A-site aminoacyl-tRNA and its amino acid identity on stalling in conjunction with the structural dynamics of the "A-site crevice" (U2506, C2452, G2505) and its modulation by erythromycin binding. The manuscript is well written easy accessible and the findings reported provide important insight into the structural dynamics that interacting small molecular inhibitors can exploit to achieve their effect on the ribosome. The observations reported constitute important contributions to our understanding of erythromycin induced stalling on the ErmBL peptide and as such should be published in Nature Communications after addressing the mostly minor concerns below.

1. Recently, another report has been published in Nature Chemical Biology (Gupta et al.: Nascent peptide assists the ribosome in recognizing chemically distinct small molecules, DOI: 10.1038/NCHEMBIO.1998, some of the authors were also co-authors on the 2014 report by Arenz et al. in Nature communications reporting a lower resolution complex) also investigating the erythromycin stalled ermBL peptide complex using MD simulations to understand the molecular mechanism of stalling by the antibiotic and ErmBL peptide. The information reported in this report (Gupta et al.) complement the observations in the manuscript under review and should be discussed in the light of the model and data reported.

Indeed, the manuscript of Gupta et al is extremely interesting and we agree that it is appropriate to discuss these findings in the light of our model and data. We have now incorporated this paper (new reference 18) into the introduction on page 4, as well as in the Conclusion on page 16.

Minor issues:

P2: Abstract, Streptococcus sanguis is not italicized.

Corrected

P4: The description about compensatory mutations in the 23S rRNA alongside the ErmBL-R7A mutation mention that the U2586 nucleotides compromised stalling efficiency, but then a point is made that the A2062 mutation does not. It is confusing what the relevance of which mutation compensates and which do not at this point in the manuscript without any context to the positions of those nucleotides in the 23S rRNA. Perhaps just mentioning the location and importance of these nucleotides in the introduction might alert the reader to their importance later in the manuscript.

We have now extended the sentence on page 4 to inform the reader in the introduction as to the location and importance of U2586 and A2062.

P7 and 8: "The resulting simulations were very stable, with root mean square deviations (rmsd)..." its not clear what is meant by "very stable". The authors should refrain from the use of this qualifier unless they provide a scale that helps assess the claim. e.g. in energy or % present over simulation time.

The reviewer is right in pointing out that here we were a bit unclear. What we meant was that the structure of the ribosome in our simulations did not change more than expected from the purely thermal fluctuations. We also wanted to emphasize that, even on the microsecond timescale of our simulations, such fluctuations are contained within an rmsd of 5.2 Å. This value is low compared to the 6-10 Å range observed in previously published, shorter simulations (30-100 ns) of ribosomes started from high resolution X-ray structures (references 29 and 34). This underscores the quality of our simulation setup and force field, as well as that of the cryo-EM starting structure. We have now clarified this in the main text and, to enable the reader to assess the quality of the simulations, we now also provide

references (and rmsd values) to previously published MD simulations of ribosomes, which deviate more from their starting structure.

- The sentence on page 7: "The resulting simulations were very stable, with root mean square deviations (rmsd) from the cryo-EM structure remaining below 5.2 Å (**Supplementary Fig. 5**, total simulation time ~12 μs)" now reads: "The root mean square deviations (rmsd) of the simulations from the cryo-EM structure remained low (rmsd <5.2 Å, **Supplementary Fig. 5**, total simulation time ~12 μs) in comparison to other, shorter MD simulations of ribosomes started from high resolution X-ray structures (rmsd 6-10 Å)^{29,34}. This underscores the quality of our simulation setup and force field as well as that of the starting structure."

P8: When describing the flexibility of the N-terminal residues during the course of the simulation it is referred to as "the first four N-terminal residues" which is confusing when this is referring to residues 2-5 that are observable in the cyro-EM structure used for the simulation not the actual first 1-4 amino acids of the ErmBL peptide.

The revised version now reads: "Removal of Ery leads to increased dynamics and flexibility of the ErmBL nascent chain, in particular the N-terminal residues 2-5 of the ErmBL peptide (Fig. 2d), as seen by the root mean square fluctuations (rmsf, Fig. 2e)."

P8: "The absence of the drug... (Fig. 2e,f)" Could 2d not also be cited here?

Yes, we now cite Fig. 2d-f

P12: Paragraph starting "The most recent model for peptide bond formation,..." describes a proton wire model of peptide bond formation yet the description does not match Fig. 6h. Specifically, it is mentioned that water 1 (W1) accepts a proton transferred by A76 of the P-site tRNA and A2451 of the 23S rRNA, but neither of these residues is shown in Fig. 6h.

The reviewer is quite correct that Fig. 6h does not adequately show what is mentioned in the text. The problem is that not all aspects can be shown in a single view. Therefore, we have now added an additional Supplementary Fig. 8 to rectify this situation.

Also in this paragraph the acronym OP1 is used, followed by O1P later the paragraph, and nowhere is it mentioned what these acronyms are in reference to.

To make this clear OP1 is used exclusively to describe a non-bridging phosphate-oxygen, and is defined in the text on page 13 and labeled in the new Supplementary Fig. S8.

Finally, also in this paragraph, the sentence "Such simultaneous interaction of L27 with A- and P-tRNA..." mentions that in the ErmBL-SRC a simultaneous interaction of L27 to both the A- and P-tRNA is not possible, but the figure referenced to show this (Fig. 6h) does not adequately portray this as the A-site tRNA is not shown. It can be deduced from previous

figures and text that the A-tRNA is far enough away that L27 does not make simultaneous interaction with both tRNAs but Fig.6h. does not show this.

The new Supplementary Fig. 8 now rectifies this situation by including simultaneously A-tRNA and P-tRNA as well as L27. Unfortunately, this was not possible in Fig 6h because of the angle needed to emphasis the absence of density for A2062 and L27.

Figure 6h - The figure legend and/or description in the text of this figure sub panel does not provide any information about the proton wire model past water 1 (W1), specifically that the hydrogen bonds to W2 and W3 are not known as indicated by the (?) on the figure. The sub panel shows the proton wire between L27 and A2602 as the authors mention in the text. At least some more detail explaining the used symbols would help.

We have now removed W2 and W3 from the figure to simplify everything and to avoid confusing the reader.

Supplemental Fig. 6 needs a more descriptive figure caption to adequately describe the data being presented in the figure for those less experienced in bioinformatics analyses of MD data. Also highlighting rows of significance that are referenced in the text allows the reader to quickly find the information.

This point was also raised by reviewer #4. In order to clarify this we have: 1) introduced the average occupancy over the whole simulation time as a metric to describe the stability of the contacts in the main text; 2) increased the readability of Supplemental Figure 6 by highlighting the contacts discussed in the main text; 3) provided a more descriptive caption for the Supplemental Figure.

Main text page 8:

All these interactions were also observed throughout the MD simulations with erythromycin; In particular the hydrogen bonds between Asp10 and G2061 were very stable in all the simulations (62% average occupancy, Supplementary Fig. 6), with the formation of additional hydrogen bonds also being seen between the amide group of Asp10 and 23S rRNA nucleotide C2063 (74% average occupancy, Supplementary Fig. 6).

Main text page 9:

Specifically, two hydrogen bonds are possible with the two terminal amino groups of Arg7, namely to the phosphate oxygen atoms of C2063 and U2441 (Fig. 3d), which were also observed in the MD simulations (with average occupancies of 13% and 87%, respectively, Supplementary Fig. 6).

Caption Supplementary Fig. 6a,b:

Hydrogen bond occupancies as a function of time for selected hydrogen bonds. Each grey scale block represents the fractional occupancy of the given hydrogen bond calculated over a 20 ns window extracted from the simulations with erythromycin (a, b). Contacts discussed in the main text are highlighted in yellow.

Caption Supplementary Fig. 6c:

(c) Hydrogen bond occupancies as a function of time for selected hydrogen bonds. Each grey scale block represents the fractional occupancy of the given hydrogen bond calculated over a 20 ns window extracted from the simulations without erythromycin. Contacts discussed in the main text are highlighted in yellow.

P13: "conformation of U2504-U2506 stayed close to that observed in the cryo-EM structure": the authors should specify what that means e.g. in RMSD or just distance.

We've added RMSD values relative to the cryo-EM structure in the text on page 13-14.

The revised version on page 13-14 now reads:

"In our study, principal component analysis (PCA, see Methods) was performed on the backbone atoms to capture their collective motion, which revealed that, in presence of erythromycin, the conformation of U2504-U2506 stayed close to that observed in the cryo-EM structure (with an average rmsd over the simulation of 1.3 Å, red trace in Fig. 7a, see also Supplementary Fig. 9). In contrast, in the simulation performed without erythromycin (green trace in Fig. 7a), these nucleotides markedly depart from the conformations observed in the simulations with erythromycin, and after ~1.5 μs, these nucleotides attained a new stable conformation (average rmsd: 2.3 Å).

P15: typo: Sentence "To test this hypothesis,..." aminoacids should read amino acids

Corrected

Reviewer #4 (Remarks to the Author):

In this manuscript, the authors employ a combination of cryo-EM and simulations to elucidate the mechanism of ErmBL-mediated translation arrest. The manuscript is written very well. A strength of the manuscript is that the simulations employed are very long (microseconds). As far as I know, there is only simulation of a full ribosome that has exceeded one microsecond, which highlights the dedicated effort the authors have made to tackling this problem. In summary, the authors use cryo-EM reconstructions to construct atomic models. These models are then used as the starting point for simulations. Together, the authors provide compelling evidence that the primary role of Erm is to impose a steric constriction on the nascent protein, which results in distortion of the CCA end of the PTC. This is proposed as the mechanism of stalling.

There are a few technical points and phrasing clarifications that are necessary before publication.

1) A few of the claims are overstated. For example, on page 5 is it stated that "MD simulations reveal that the presence of erythromycin also influences the position of the aminoacyl moiety of the A-tRNA via a cascade

of 23S rRNA rearrangements." This is a prediction of the simulations, it is not shown to be true.

We have changed the word "reveal" to "predict" in this sentence.

2) A major claim of this study is that Erm induces a distortion in the CCA end. However, the reported simulations are several microseconds. There should be more complete analysis showing that there isn't some other effect that is causing the distortion. For example, in another microsecond explicit-solvent simulation of the ribosome (Whitford et al PLOS Comp Bio 2013), it was shown that the subunits can rotate several degrees. The author do state that the RMSD is 5.2Å, but this sounds like a large value. What are the rotation angles over that course of time? It is possible that rotations could strain the tRNA molecules, which would also contribute to distortions. Also, where were the largest changes in conformation (large RMSD per residue)? Since there are hundreds of thousands of atoms, an average of 5.2 could still involve extremely large deviations of some regions.

As we understand, the reviewer assumes that the mentioned CCA distortion is a simulation result. However, the distortion is observed already in the cryo-EM, we think the reviewer was misled by our introductory statement on pages 4 and 5:

This has now been modified to clarify that this distortion is indeed seen already in the cryo-EM structure:

"In the cryo-EM structure, the unusual conformation of ErmBL appears to distort the terminal A76 ribose of the P-tRNA, which is supported by MD simulations performed after removal of erythromycin showing that the ribose distortion is alleviated as the ErmBL nascent chain moves into the volume previously occupied by the drug."

It should also be noted that in the experimentally derived structure, no anomalous rotation of the two subunits was observed when comparing with other ribosome structures of the same state in the absence of erythromycin. We have addressed the other questions of the reviewer by calculating the intersubunit rotation angles in the course of all the simulations and comparing the results to those previously published by Whitford et al.

As can be seen from the resulting plot (see above), the rotation angles visited by the simulations span the same range of values as those seen by Whitford et al. ($\sim 5^\circ$). The traces in presence and absence of the antibiotic are qualitatively indistinguishable. Further, we checked if the regions undergoing the largest conformational changes were outside of the region of interest, namely, the surroundings of the CCA-ends of the tRNAs, the attached A-site amino acid, and the P-site peptide. Indeed, the rmsd of the atoms within a range of 25 Å was contained within ~ 2 Å. The time traces of this rmsd value for each simulation have been included in Supplementary Figure 5.

The caption of Supplementary Figure 5 now reads:

“Root mean square deviation (rmsd) as a function of simulation time for all simulations calculated for the entire ribosome (colored traces) as well as for a subset of atoms within 25 Å of the PTC (Black and grey traces). Top panel: with erythromycin (WT: red, dark red, black and dark grey; Lys11Ala mutation: orange and light grey). Central panel: with erythromycin starting from perturbed Asp10 conformation (WT: magenta, dark magenta, black and dark grey; Lys11Ala: blue and light grey). Bottom panel: without erythromycin (WT: green, dark green, black and dark grey; Lys11Ala: cyan and light grey). In addition, the rmsd from the starting point is shown for the simulations including the Lys11Ala mutation (lower curves in orange, blue and cyan, respectively).”

3) On page 8, it is claimed that hydrogen bonds between Asp 10 and G2061 were very stable in all simulations. While the authors did perform long simulations, by computational standards, showing that hydrogen bonds are formed for a few microseconds does not mean they are "very stable". This claim should be presented in a quantitative fashion. For example, for what fraction of the simulated time were these hydrogen bonds formed?

See reply to reviewer #3

4) Related to point 3: On page 9, it is stated that hydrogen bonds were observed between C2063 and U2441. Again, a fraction of time formed would be helpful. Figure S6 is not easy to interpret.

Also here see reviewer #3, we have now added fractional occupancies for all hydrogen bonds discussed in the text.

5) On page 9, it is stated that " mutation of U2586 were able to restore stalling" Since this is referring to a previous publication, it should probably be "is able to restore stalling". As written, it sounds like it is a finding from the current study.

We have replaced this text with "were shown previously to restore stalling"

6) On pages 11 and 12, there are multiple claims of shifts in positions. Since there is no absolute reference frame for comparison, it is not clear what exactly these values signify. These atoms move relative to what?

All the shifts are relative between the Lys-tRNA and either the accommodated or unaccommodated tRNAs. The reviewer is correct in that the reference frame is

based on a complete alignment of the 23S rRNA between the 50S subunits of the ErmBL-SRC and the relevant X-ray structures, in this case PDBs 1vq6, 4qcm and 1vqn. As requested also by reviewer #1 (point 3) we have now described how the alignments were performed in the methods and provided the rmsds.

7) On page 14, it is stated "Given that peptide bond formation takes place on timescales much shorter than those in the simulations, it is the smallest d_2 values observed in the simulations that must be taken into account" According to biochemical data, peptide bond formation can be the rate limiting step (Molecular Cell 30, 589-598, 2008), and rates of 250,000/s (corresponding to the timescale of the simulation) have not been reported. Please clarify this statement.

We thank the reviewer for this comment. We have realized that the sentence is potentially misleading given the accepted use of the term "peptide bond formation" in the field. This standard definition considers the total peptide bond formation rate as the rate of transitioning from the pre-attack state (with a fully accommodated A-site tRNA) to the post-attack state (k_{pt} in Molecular Cell 30, 589-598, 2008). This rate can be considered as the product of two contributions: the rate of attacking the barrier from the pre attack minimum multiplied by the rate of the actual chemical step. It is this latter rate that, in our argument, we assume to be fast in comparison to the simulation timescale. This assumption is now explicitly laid out in the text. If this chemical step is fast compared to the attack rate, the attack rate is rate limiting for the k_{pt} . It is true that from our simulations we cannot access the attack rate due to limited sampling. In our argument, we additionally assume that, given two simulated systems, the one that reaches configurations closest to the conformation necessary for the fast chemical step to occur has a faster attack rate.

The fact that our simulations without erythromycin (or with the K11A mutation) reach configurations closer to the conformation where the chemical step is possible (d_2 in fig.7) indicates that the attack rate is faster in these simulations.

In the revised text the sentence on page 14 has been changed to: "We assume that the rate of the chemical step of peptide bond formation is faster than the rate of reaching conformations where this chemical step is possible. Therefore, simulations that more closely approach these conformations (i.e., smallest d_2 values) are assumed to represent the complexes with the fastest peptide bond formation rates."

8) Page 14: "To test this prediction" The previous statement is not a prediction, it is notion/idea/suggestion/rationalization. Since the current paper has actual predictions, it is confusing to use the term in this context. Following this, it is stated "As predicted, the strongest interactions...." This is very misleading. That is, within in this statement, a hypothesis is treated as a prediction, and a simulation is treated as a test of the prediction. A more appropriate phrasing would be "As expected/suggested, the model predicts that..."

We agree in the sense that we consider an idea that has a low (prior) probability as "just an idea", on page 14, this now reads: To test this idea...

We think the reviewer will also agree that an idea which has a very high probability is a prediction. We argue that the test we describe here increases the probability to a high degree so we think of it as a prediction, and would rather keep the wording of the second part as in the original manuscript.

9) Similarly to the previous point, on page 15, it is stated that "much weaker interaction enthalpies were observed when Lys11..." Since this manuscript provides a combined description of experiments and computation, in order to clearly distinguish between theoretical and experimental results, the computational results should be described as predictions and experiments should be reported as observations.

On this point, which is mostly about careful wording, we respectfully disagree with the reviewer: indeed if we were to use this wording, we would have to revert from "idea" as correctly suggested by the reviewer in point 8, back to "prediction". In fact, we don't think every computational result is a prediction, as in the case mentioned above. We prefer to keep referring to computational results as "observed in the simulation", which avoids this misinterpretation.

Reviewers' Comments:

Reviewer #1 (Remarks to the Author)

The authors have addressed my concerns, and I believe that the manuscript is ready for publication on Nat. Comm.

Reviewer #2 (Remarks to the Author)

The authors have appropriately responded to the request with respect to the over-fitting and half-sets questions, providing a cross-FSC statistic that, while not completely free, at least provides some measure of independent validation of the model, and a clear disclaimer stating that their half-sets were not refined independently. For future reference, it is important to note that this refinement technique would be extremely suspect if the obtained resolution would be much closer to the cut-off used for refinement. In other words this technique is not acceptable unless the reported resolution is far better than what was used in the initial filtering so that the agreement in the FSC curve extends clearly beyond the resolution imposed in the original filter. On a related note, reviewer 1 insists that the authors should use particular software, this should be left to the authors unless the software used is clearly inappropriate for the task it is used to perform, which is not the case here. It is not the place of the reviewers or the journal to dictate the choice of software, or refinement technique, unless there is a clear problem with the results - and as mentioned previously, this would not appear to be the case.

However, on the issue of the proposed simulations, the authors have not appropriately revised the manuscript. While it is possible that the simulations require substantial investments of computational time, it is not acceptable to present the results of a single, selected mutational simulation. The author's statement that the mutation they present was "carefully chosen" does not encourage confidence in this respect. The current approach with simulation for only one particular mutation is insufficient since it does not provide any validation, statistical justification or proper control. Validating the results obtained through simulations is particularly important considering that the presented structural data is only marginally better compared to the previously published structure and therefore simulations constitute a very important component of the manuscript.

Reviewer #3 (Remarks to the Author)

This revised version of the manuscript "A combined cryo-EM and molecular dynamics approach reveals the mechanism of ErmBL-mediated translation arrest" by Stefan Arenz and co-workers describes a new 3.6A-resolution cryo-EM structure of the ErmBL stalled ribosome complex and subsequent molecular dynamic simulations using the stalled complex.

As indicated in the original review, the manuscript provides interesting insight into the structural dynamics of ErmBL induced stalling of ribosome dependent protein synthesis, including a improved model for the observed sequence-dependence of ErmBL induced stalling.

All concerns raised have been addressed appropriately. In particular the findings from recent related studies (Gupta et al.: Nascent peptide assists the ribosome in recognizing chemically distinct small molecules) are now incorporated, demonstrating the significance of the work reported in the manuscript by Arenz and co-workers. The use of qualifying terms also raised by other reviewers has been addressed.

I am looking forward to seeing the work published.

Reviewer #4 (Remarks to the Author)

All queries have been sufficiently addressed. Their modifications to the manuscript have resulted in a more clear distinction between which findings are from experiments, and which are from simulations.

While the simulations, alone, do not constitute a major advance (although reaching multiple microseconds is an impressive milestone), I disagree with the assertion by reviewer 2 that the current simulations require additional validation, or that the claims are overstated. That is, the simulations of the mutated system do provide evidence that the CCA end can more fully accommodate when the amino acid is mutated to Ala. This observation is consistent with the working notion that by allowing more complete accommodation, mutants may bypass stalling. I agree that additional simulations could be performed (e.g. of a subset of atoms), though I'm not sure exactly what one would expect to learn that is not already implicated by the current simulations. These simulations do provide evidence in support of a mechanistic interpretation that relates Erm and stalling. In my opinion, the authors don't make major quantitative claims that would require more significant statistics, and there are already signatures in their data. Within a broader context, I think the simulations represent a good step forward for the field. That is, it is far too common for groups to report the dynamics of 10-100ns and then make sweeping claims. By pushing their simulations into the microsecond regime for a large system, there will be positive pressure on the field to increase the resources dedicated to each calculation. While that technical aspect would not be sufficient grounds for a major paper, it is certainly a positive feature of the overall manuscript.